# A hierarchical atlas of the human cerebellum for functional precision mapping

Caroline Nettekoven [1,2] ✉, Da Zhi [1,2], Ladan Shahshahani[1], Ana Luísa Pinho [1,2], Noam Saadon-Grosman[3], Randy Lee Buckner [3] & Jörn Diedrichsen [1,2,4] ✉

The human cerebellum is activated by a wide variety of cognitive and motor tasks. Previous functional atlases have relied on single task-based or resting-state fMRI datasets. Here, we present a functional atlas that integrates information from seven large-scale datasets, outperforming existing group atlases. The atlas has three further advantages. First, the atlas allows for precision mapping in individuals: the integration of the probabilistic group atlas with an individual localizer scan results in a marked improvement in prediction of individual boundaries. Second, we provide both asymmetric and symmetric versions of the atlas. The symmetric version, which is obtained by constraining the boundaries to be the same across hemispheres, is especially useful in studying functional lateralization. Finally, the regions are hierarchically organized across three levels, allowing analyses at the appropriate level of granularity. Overall, the present atlas is an important resource for the study of the interdigitated functional organization of the human cerebellum in health and disease.

Decades of neuroimaging have shown cerebellar activation in a broad range of tasks, including motor, social, and cognitive tasks—yet its contribution to these different functions remains elusive[1,2]. A major obstacle to understanding the cerebellar contribution is that the cerebellum consists of a mosaic of functional regions, specialized for distinct roles[3]. It is still common to use the anatomical subdivision into different lobules[4,5] to define regions of interest, even though lobular boundaries do not align with boundaries in functional specialization[3].

There are several existing maps based on resting-state or task-based functional Magnetic Resonance Imaging (fMRI) data[3,6,7] that parcellate the cerebellum into functional regions. These functional atlases outperform anatomical parcellations at predicting functional boundaries on an independent task set, with a task-based parcellation based on a large multi-domain task battery (MDTB) being particularly powerful[3]. Nonetheless, parcellations based on single datasets usually show some distinct weaknesses: For example, the MDTB parcellation[3] does not delineate the foot or mouth motor region very well, likely because of the absence of those movement types from the task set. Any single dataset and analysis approach will necessarily emphasize some features over others. To address these shortcomings, we have recently developed a Bayesian Hierarchical method that combines information across datasets into a single parcellation[8]. In this study, we apply this model to seven large task-based datasets to derive a comprehensive cerebellar functional atlas.

Another important limitation of existing group atlases is that they ignore the large inter-individual variability in functional brain organization[9–13]. This problem is particularly relevant for the cerebellar cortex, where many functionally heterogeneous regions are packed into a relatively small volume[3,14,15]. Multiple groups have therefore pursued a precision mapping approach, using localizing data to define functional regions at the individual level[10–12,15]. To enable such precise and fine-grained analysis, the present atlas is based on a probabilistic framework, which allows the user to use even limited individual data to optimally tailor the atlas to an individual[8,16]. We evaluated this approach carefully by showing the utility of the personalized parcellation at predicting boundaries and functional specialization in the same individual in different tasks, as compared to both the group atlas, and a parcellation solely based on individual data.

[1]Western Institute for Neuroscience, Western University, London, ON, Canada. [2]Department of Computer Science, Western University, London, ON, Canada. [3]Harvard University, Cambridge, USA. [4]Department of Statistical and Actuarial Sciences, Western University, London, ON, Canada. ✉e-mail: cr.nettekoven@gmail.com; jdiedric@uwo.ca

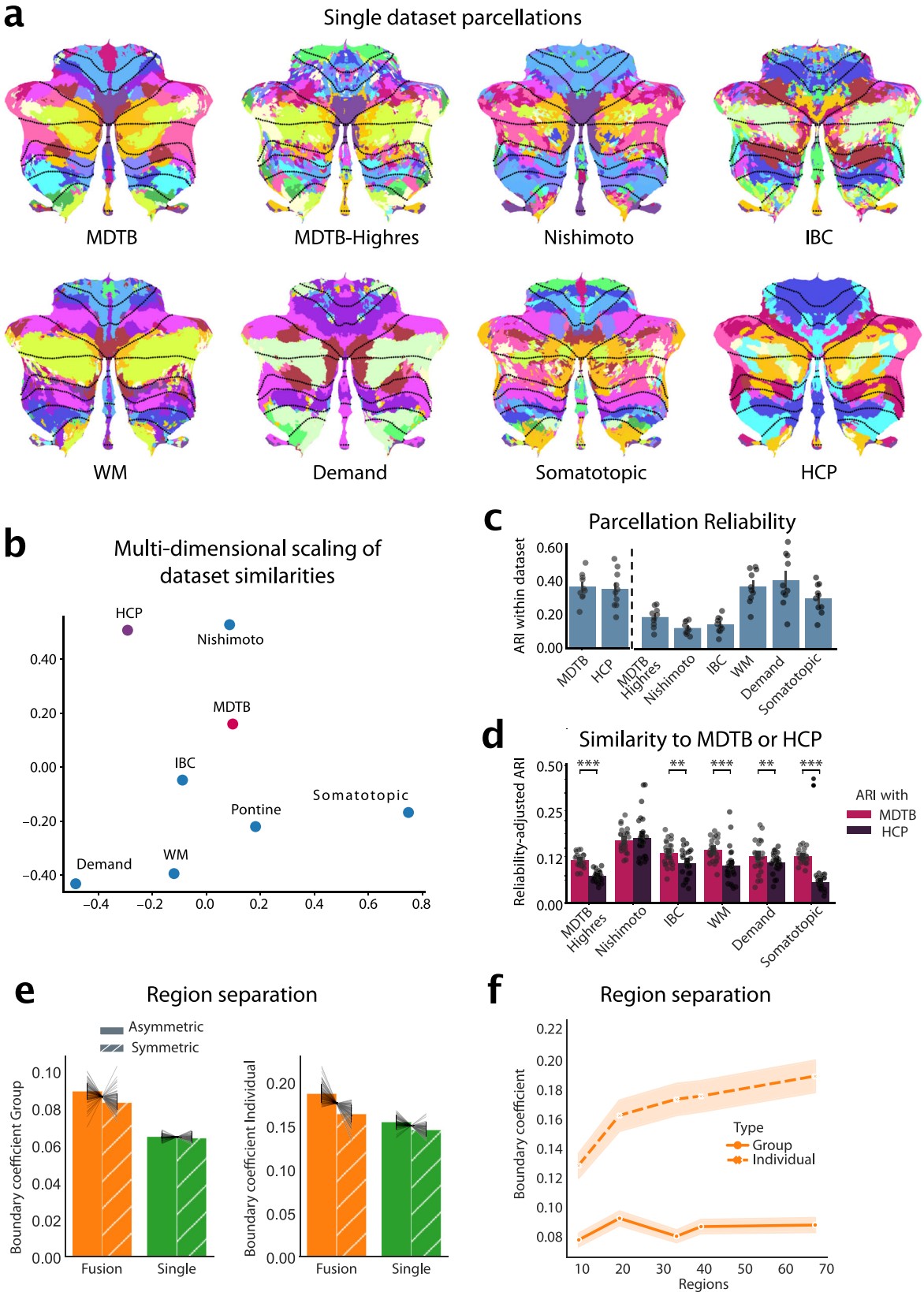

The cerebellum plays a key role in lateralized functions (i.e., language[17]) and shows lateralized developmental trajectories[18]. The study of lateralization, however, is complicated by existing functional atlases, as they have asymmetric boundaries with ambiguities in correspondence between between left and right regions. We therefore developed a version of the atlas with symmetric boundaries and

matching hemispheric parcel pairs. Importantly, we did not constrain the functional profiles to be the same across hemispheres, enabling us to study functional lateralization. The comparison to an asymmetric version of the atlas also allowed us to assess whether this symmetry constraint is adequate, or to what degree the spatial organization is truly asymmetric.

**Fig. 1 | Building a functional atlas of the cerebellum across datasets.**
**a**, Parcellations (K = 68) derived from each single dataset (multi-domain task battery dataset, MDTB; high-resolution multi-domain task battery dataset, Highres-MDTB; Nishimoto dataset, individual brain charting dataset, IBC; working memory dataset, WM; demand dataset, Demand; somatotopic dataset, Somatotopic; Human Connectome Project dataset, HCP). The probabilistic parcellation is shown as a winner-take-all projection onto a flattened representation of the cerebellum[32]. Functionally similar regions are colored similarly within a parcellation (see methods: parcel similarity) and spatially similar parcels are assigned similar colors across parcellations. Dotted lines indicate lobular boundaries. **b**, Projection of the between-dataset adjusted Rand Index (ARI) of single-dataset parcellations into a 2d-space through multi-dimensional scaling (see methods: Single-dataset parcellations and similarity analysis of parcellations). **c**, Within-dataset reliability of parcellation, calculated as the mean ARI across the 5 levels of granularity (10, 20, 34, 40 and 68 regions). Errorbars indicate SE of the mean across the five granularity pairs. Dots show individual reliability values (n = 10). **d**, Reliability-adjusted ARI between each single-dataset parcellations and the multi-domain task battery (MDTB; task-

based) and Human Connectome Project (HCP; resting parcellation) parcellation. Errorbars indicate standard error of the mean across the five levels of granularity. Dots show individual similarity values (n = 25). Paired two-tailed t-tests were calculated between the ARI of each single-dataset to the MDTB parcellation and to the HCP parcellation at each granularity: MDTB-Highres: $t_{24} = 16.404, p = 1.523 \times 10^{-14}$; IBC: $t_{24} = 3.513, p = .0017$; WM: $t_{24} = 4.727, p = 8.318 \times 10^{-5}$; Demand: $t_{24} = 3.262, p = .0033$; Somatotopic: $t_{24} = 12.538, p = 5.015 \times 10^{-12}$. ** $p<0.01$, *** $p<0.0001$. **e**, Distance-Controlled Boundary Coefficient (DCBC) evaluation of the symmetric and asymmetric atlas averaged across granularities evaluated on the group map (left) or on individual maps derived with that atlas (right). Errorbars indicate SE of the mean across subjects. Gray connecting lines show individual subjects (n = 111). For visualization purposes of the subject data, the subject mean was subtracted and the group mean added. **f**, DCBC evaluation of the symmetric group map and of individual maps derived from the model with 10, 20, 34, 40, and 68 regions. Shaded area indicates SE of the mean across subjects. Source data are provided as a Source Data file.

---

Finally, questions about cerebellar function will benefit from being tested at different levels of granularity. For many anatomical and patient studies, it is often most appropriate to summarize measures in terms of broad functional domains (e.g., motor vs. social-linguistic-spatial regions), whereas more detailed functional studies require the definition of finer region distinctions (e.g., separate hand, foot, and tongue regions within the motor domain or separation between social and linguistic domains). We therefore created the atlas with a hierarchical organization of functional regions where the boundaries of the broad domains remain the same at each level of granularity.

## Results

### Different fMRI datasets reveal a similar, but not identical, cerebellar organization

A common functional atlas across different datasets only makes sense, if we assume that there is a robust functional organization that remains the same across tasks. However, the cognitive state of the brain (rest or specific tasks) likely influences how different functional regions work together. Therefore, parcellations based on different datasets may highlight different functional boundaries. As a first step, we therefore sought to characterize similarities between parcellations based on single datasets, using task-based and resting-state data. We trained our probabilistic parcellation model[8] on seven task-based and one resting-state datasets (Supplemental Table. 1) in isolation and then compared the resultant parcellations (Fig. 1a).

The parcellations overall showed clear similarities, but also some dataset-specific differences. A smooth boundary between motor regions in lobule I-VI and cognitive regions in lobule VII was present in all parcellations (e.g. between the magenta and pink regions in MDTB and Demand dataset in lobule VI). On the other hand, the ability to distinguish regions within motor and cognitive regions differed between datasets. For example, the somatotopic dataset only tested individual body movements, and therefore resulted in a clear somatomotor map, but did not delineate cognitive regions in lobule VII well, as can be seen by the fragmented pattern in Crus I/II and lobule IX. In contrast, the Demand dataset delineated regions involved in working memory and executive functions, but did not lead to a clear somatomotor map. Parcellations based on resting-state data (HCP) showed consistent boundaries in regions related to the default network (lobules VII) but appear to delineate other regions (e.g. motor) less finely.

To quantify these similarities, we calculated the adjusted Rand Index (ARI) between parcellations at different levels of parcel granularity (10, 20, 34, 40 and 68 regions). The indices were averaged across granularities and normalized by the within-dataset ARI (Fig. 1c, see methods). Overall, the resultant reliability-adjusted ARIs were positive across all dataset pairs (One-sample t-test of the between-dataset ARIs

averaged across granularities $t_{27} = 17.885, p = 1.696 \times 10^{-16}$), indicating that there are clear commonalities across all different task and resting state datasets[10,19,20].

To assess the similarity of the resulting parcellations better, we visualized the reliability-adjusted ARIs using multi-dimensional scaling (Fig. 1b). Unsurprisingly, task-based datasets that test similar task domains (i.e., working memory and multi-demand dataset) resulted in similar parcellations. The Somatotopic and the resting-state (HCP) parcellation occupied two other, opposing poles in the space of parcellations.

Parcellations based on datasets that included a large range of cognitive tasks (MDTB, MDTB-Highres, and IBC) occupied a middle position, suggesting that such parcellations can well capture stable features of functional boundaries across tasks. Indeed, when we compared the ARI for each specific task-based parcellations, we found that they were more similar to the parcellation derived from the MDTB dataset than to one derived from the HCP dataset (paired t-test: $t_{149} = 9.605, p = 2.672 \times 10^{-17}$; Fig. 1d). Testing each set of task-based parcellations separately confirmed that all, except for the Nishimoto parcellations ($t_{24} = -0.838, p = 0.410$) were significantly more similar to the MDTB than the HCP (resting-state) parcellations (MDTB-Highres: $t_{24} = 16.404, p = 1.523 \times 10^{-14}$; IBC: $t_{24} = 3.513, p = .0017$; WM: $t_{24} = 4.727, p = 8.318 \times 10^{-5}$; Demand: $t_{24} = 3.262, p = .0033$; Somatotopic: $t_{24} = 12.538, p = 5.015 \times 10^{-12}$). As indicated by the opposing poles occupied by Somatotopic dataset and HCP resting-state dataset (Fig. 1b), this difference was largest for the Somatotopic dataset, suggesting that rest and single-limb movements reveal quite dissimilar boundaries.

In sum, this analysis shows that the resting-state parcellation captures many task-based boundaries, but also differs from a parcellation that delineates somatotopic motor regions. This is in line with previous observations that resting-state data do not always reveal motor regions of the cerebellum clearly[7,21]. In practice we found that the inclusion of resting-state data into the fused atlas tended to prevent a clear delineation of somatomotor regions. For the final atlas we therefore decided to rely on task-based data only given the goal here of comprehensively mapping motor and non-motor cerebellar regions.

### Dataset fusion improves prediction of functional boundaries

Our Hierarchical Bayesian Parcellation framework[8] allows for data fusion by modeling each dataset separately and then combines them iteratively into a common group atlas. In this process, each dataset is weighted by a measure of its reliability (see methods, Hierarchical Bayesian parcellation framework).

To verify that the fusion of datasets through our framework systematically improved on single-dataset parcellations, we adopted a leave-one-dataset-out approach. We trained the fusion parcellation on

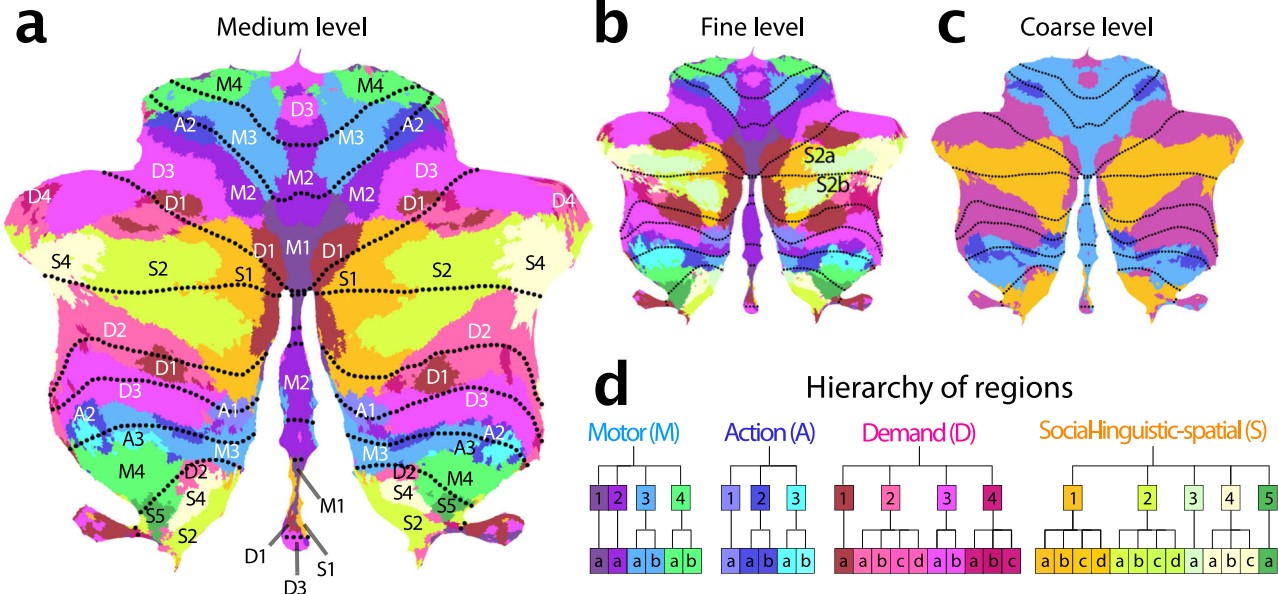

**Fig. 2 | Cerebellar functional atlas at three levels of granularity. a** Medium granularity with 32 regions; 16 per hemisphere. The colormap represents the functional similarity of different regions (see methods: parcel similarity and clustering). **b** Fine granularity with 68 regions; 34 per hemisphere. **c** Coarse granularity with four functional domains. The symmetric version of the atlas is shown, for the asymmetric version, see Fig. 4. **d** Hierarchical organization based on the functional similarity of regions, depicted as a dendrogram. The label of each region indicates the functional domain (M,A,D,S), followed by a region number (1-4), and a lowercase letter for the subregion (**a**–**d**).

all task-based datasets except one and tested its ability to predict the functional boundaries within that left-out dataset. This ability was quantified using the Distance-Controlled Boundary Coefficient (DCBC) which compares the correlation between within-parcel voxel-pairs to the correlation between voxels-pairs across a boundary, while controlling for spatial distance[22], with higher values indicating better performance. We found that the fused group atlas outperformed single dataset parcellations averaged across granularities ($t_{110} = -4.466, p = 1.936 \times 10^{-5}$; Fig. 1e left).

In addition to providing a winner-take all group map, our framework can also provide individual parcellations by integrating subject-specific data (see methods: individual precision mapping). This ability critically depends on the group atlas not only having appropriate boundaries, but also quantifying the uncertainty across participants adequately. We found that individual parcellations based on the fused atlas outperformed those derived from single dataset ($t_{110} = -2.564, p = .0171$; Fig. 1e right), confirming the superiority of the fused atlas, both when using a winner-take-all projection or a probabilistic parcellation to derive individual maps[8].

**Comparing symmetric and asymmetric atlases**

To enable the study of hemispheric specialization, we initially constrained our atlas to have spatially symmetric regions across the left and right cerebellar hemispheres, while allowing different functional profiles. To determine how much this constraint forced the group map to deviate from the true functional organization, we also estimated an asymmetric version of the atlas without using the symmetry constraint (see methods, Symmetry constraint).

We compared the ability of the asymmetric and the symmetric atlas to predict functional boundaries, again adopting a leave-one-dataset-out approach. For the group DCBC, we found a small, but significant difference between the asymmetric and symmetric atlas across levels of granularity (10-68 regions; $t_{110} = -2.344, p = .0201$) (Fig. 1b). This advantage was larger at the individual level ($t_{110} = -5.023, p = 1.981 \times 10^{-6}$). Overall, however, the predictive power of the symmetric atlas was only 5% (group) or 14% (individual) smaller than the asymmetric versions. Given the many practical uses of

the symmetric atlas for controlling for region size and location in lateralization studies, we provide both symmetric and asymmetric versions of the final atlas.

**Basemap for hierarchical atlas outperforms existing parcellations**

Instead of choosing a fixed number of regions, we used three nested levels of resolution, linked in a hierarchical scheme. This allows the user to analyze their data at different levels of granularity in a consistent fashion. To decide on the "base map" of this hierarchy, we examined the predictive performance of the fusion atlas across the tested levels of granularity at the group and individual levels (Fig. 1f). We found that the performance of the group map saturated early, reaching its best value at 20 regions. However, this peak was not significantly different from the finest granularity of 68 regions ($t_{110} = 2.783, p = .0063$). In contrast, the ability to predict boundaries in the individual increased monotonically, with the finest granularity outperforming the next lower granularity of 40 regions ($t_{110} = 7.584, p = 1.143 \times 10^{-11}$). We therefore based the hierarchical atlas on the map with the finest granularity of 68 functional regions.

The fused atlas based on all datasets significantly outperformed existing parcellations in predicting boundaries tested on all datasets. Across all subjects of all evaluation datasets, both the symmetric and the asymmetric atlas base map resulted in a higher average DCBC than existing anatomical (Lobular[5]:), task-based (MDTB[3]: and resting-state parcellations (7 and 17 regions[7]: 10 regions[6]:), all $t_{110} > 3.545, p < 5.788 \times 10^{-4}$ (see Supplemental Fig. 1)

We then clustered the 34 regions per hemisphere of the base-map into 16 regions per hemisphere according to the functional similarity between regions (see methods: parcel similarity and clustering). Finally, we organized these 16 regions into 4 broad functional domains. Based on their functional activation profiles, we denoted these four functional domains as motor (M), action (A), multi-demand (D), and social-linguistic-spatial (S) (Fig. 2c). At the medium level, we numbered the regions within each domain from medial to lateral (Fig. 2d). Finally, the finest level was annotated with a lowercase letter (a-d). In the following description of the regions,

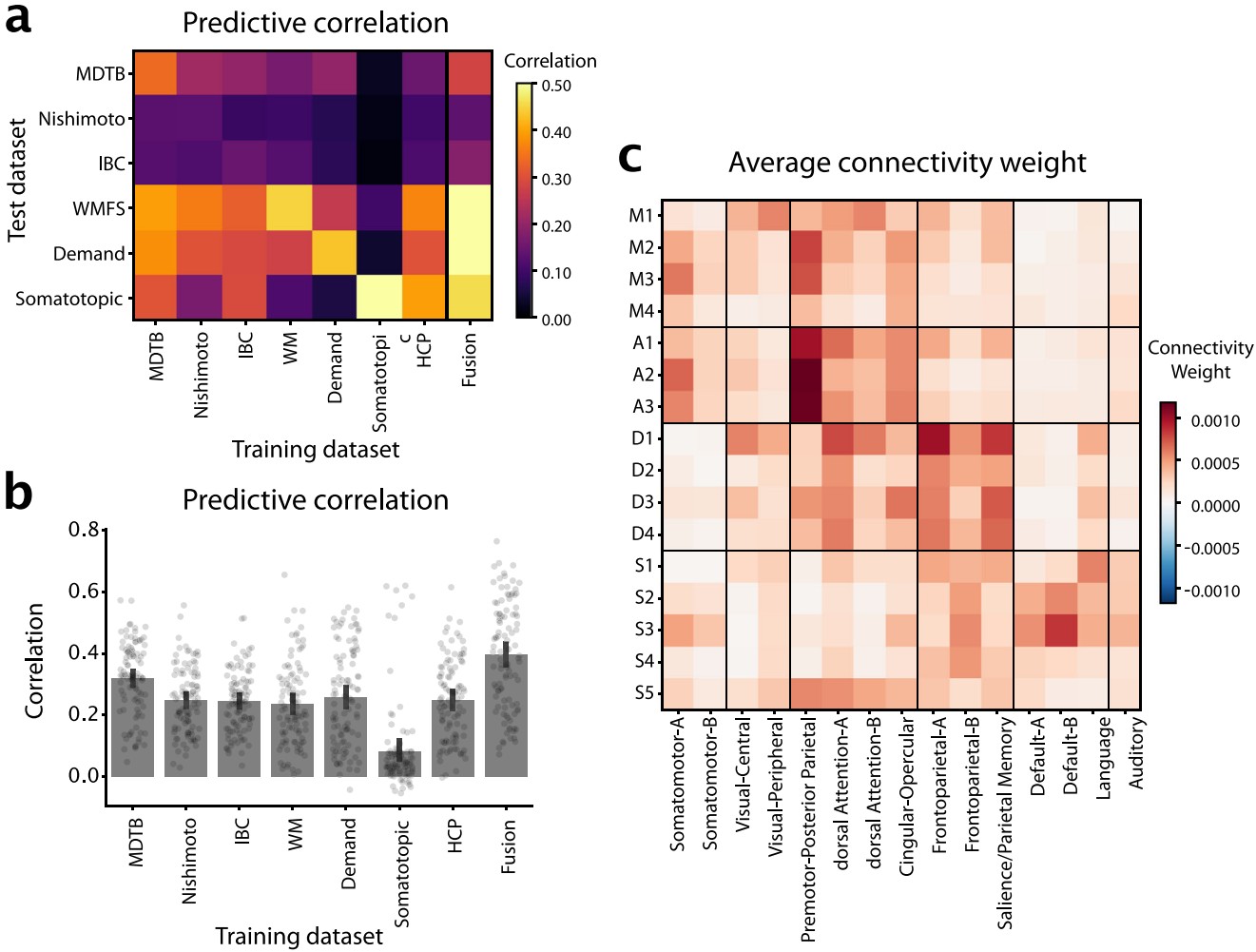

**Fig. 3 | Cerebro-cerebellar connectivity models. a** Matrix shows the correlation between observed and predicted cerebellar activity patterns for each test dataset (rows). Connectivity models were trained on each training datasets (columns) separately. Evaluation was cross-validated across subjects when training- and test-dataset were identical. **b** Correlation between observed and predicted activity patterns, averaged across test-datasets. The Fusion model used the average connectivity weights across all task-based datasets (excluding the HCP resting-state data). **c** Average connectivity weights between each cerebellar region (row), and each of the 15 resting-state networks as described in[du2023]. Source data are provided as a Source Data file.

---

we will focus on the medium level, as it provides a good compromise between precision and succinctness.

### Characterization of functional regions

Each functional region is characterized by its response profile across datasets and its spatial distribution across individuals. In describing the functional profile, we focused on responses estimated from subject-specific regions in the MDTB dataset (see methods: Functional profiles for the MDTB dataset), supplemented by more domain-specific datasets for the motor and demand regions (Somatotopic, Demand, WM).

### Motor regions

Regions that exhibited a clear preference for movements of a specific body part were grouped into the motor domain. All regions had a superior (lobules I–VI) and an inferior (lobule VIII) aspect. We also found a third representation of these body-part-specific regions in the posterior vermis, consistent with recent results at the individual subject level[21].

M1 encompassed the oculomotor vermis, which responded most strongly to saccades (Fig. 2). Even when correcting for the number of saccades, the area was further activated when participants had to read text (Theory-of-Mind), watch a movie (animated movie), or search for

visual stimuli (spatial map and visual search), likely due to the attentional demands of these tasks. Previous work has shown that this region also has a clear retinotopic organization[23]. M2 comprises a lateral and a vermal part. The lateral section showed strong responses to tongue movements in the somatotopic dataset. In contrast, the vermal component was activated by multiple different bodily movements, but otherwise was functionally most similar to the lateral M2. The M3 regions were selectively activated movement of the ipsilateral hand (Supplemental Fig. 2). Finally, M4 was most activated by movements of the lower body, including flexion and extension of the foot (Highres-MDTB), as well as contraction of the gluteal muscles (Somatotopic).

### Action regions

Directly adjacent to the motor regions lie the action regions, which were activated during action observation and motor imagery tasks. A1 and A2 both comprised spatially separate superior and inferior sections. A1 can be found medially to the hand region in lobule VI and at the border of VIIIa/VIIIb. A2 lies laterally adjacent to the superior hand region M3, and at the border of lobule VIIIa/VIIIb. In contrast, A3 primarily occupies the inferior cerebellum (Fig. 3), located at the border of lobules VIIIa/VIIIb.

Although both motor and action regions activated during movement execution, only the action regions activated when observing

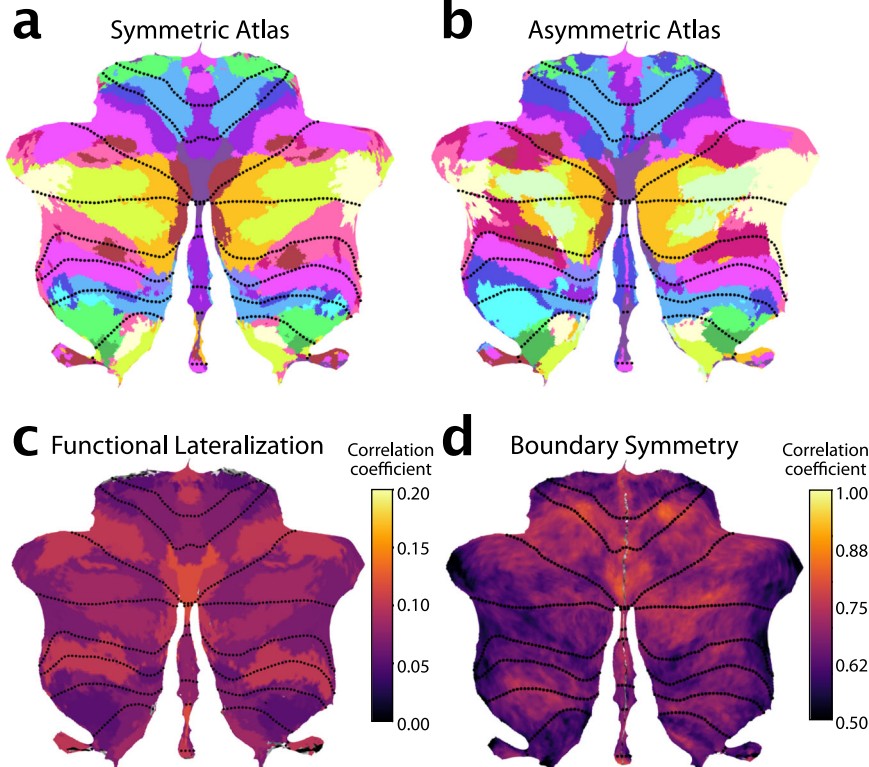

**Fig. 4 | Functional lateralization and Boundary asymmetry in the cerebellum. a** Symmetric atlas winner-take-all map. **b** Asymmetric atlas winner-take-all map. **c** Functional lateralization quantified as the correlations of the functional responses of anatomically corresponding voxel of the left and right hemisphere, averaged across subjects and within each functional region. **d** Boundary symmetry calculated as the correlations of the probabilistic voxel assignments between the symmetric and asymmetric version of the atlas.

actions without execution: In the MDTB dataset, they showed strong responses to an action observation task (video actions in Fig. 7). A1 appeared to be particularly involved where spatial simulation is required (strong responses during spatial map and mental rotation tasks). Meanwhile, A2 seems to be a classic action observation region, with little response to tasks that do not involve action observation or execution. In contrast, A3 was also activated during imagined movements (motor imagery).

### Multiple-demand regions

Tasks involving executive control, including updating, shifting and inhibition, consistently activated regions in lobules VI and VII. Based on work by Duncan et al.[24], we labeled these regions the multi-demand domain (D for short). D1 occupied the most medial portion of Crus I and II. Further out in the hemispheres, the demand region formed a "shell" around the more central social-linguistic-spatial domain (Figs. 4b, 5). Here, D3 formed the outermost layer and D2 the innermost, with D1 being interspersed between. The regions (especially D2) also had a repeated representation in lobule IX (Fig. 3). This is consistent with a 3-fold representation[7]. Intriguingly, we found also a vermal section of D3, both in lobule IV and IX. D4 was the smallest identified region. Functionally most similar to D1, it occupied the most lateral portion of the demand regions.

Consistent with the characteristics of the cortical multi-demand system[25], all regions showed significant activation during executive tasks (n-back, switch and stop tasks), and increased activity especially with high difficulty. Nonetheless, there was some functional specialization across the regions. In the MDTB dataset, D1 appeared to be involved strongly in spatial tasks, such as the mental rotation, and spatial map task. D1 and D4 were strongly engaged in the n-back task. In contrast, D2 and D3 were specifically activated by the digit span task

tested in the WM data set—with D2 more active during backwards recall and D3 showing strong increases with working memory load.

### Social-linguistic-spatial regions

The regions in hemispheric lobules Crus I and Crus II, located laterally to the D1 region, were activated by tasks involving social and linguistic processes. They also showed high activity during rest, consistent with the description of this area as the cerebellar node of the default network[7]. We identified four regions, each spanning both sides of the horizontal fissure, with S1 being the most medial and S4 most lateral (Fig. 2). S3 overlapped substantially with S2 and S4 and therefore could only be reliably differentiated from these two regions at the level of the individual (see 5a). In the volume (Supplemental Fig. 4) S1 occupies the depth of the horizontal fissure, and S4 the most lateral tips of Crus I and II. A third representation of S2 and S4 can be found in lobules IX. S1 and S2 also occupy sections in the inferior vermis (VIIIb and IX, Supplementary Fig. 4). While all regions shared some overall similarity in their response profile, there were clear inter-regional and inter-hemispheric differences. The mean evoked responses for the MDTB dataset (Supplemental Fig. 2) showed right S1 to be primarily involved in linguistic processing, with highest activation during verb generation. S2 was strongly engaged in social processing, with highest activity during a theory-of-mind task on the right and during an animated movie on the left. S2, S3, and S4 showed high levels of activity during rest. S4 and S5 appeared to be particularly involved in imagination and specific forms of self-projection (Supplementary Fig. 6a, b), showing the highest activation during the spatial and the motor imagery tasks, which require the participant to imagine themselves walking through their childhood home and playing a game of tennis, respectively. In contrast to S4, S5 was also active during a spatial working memory task (Spatial Map) and did not appear to be engaged in linguistic processes

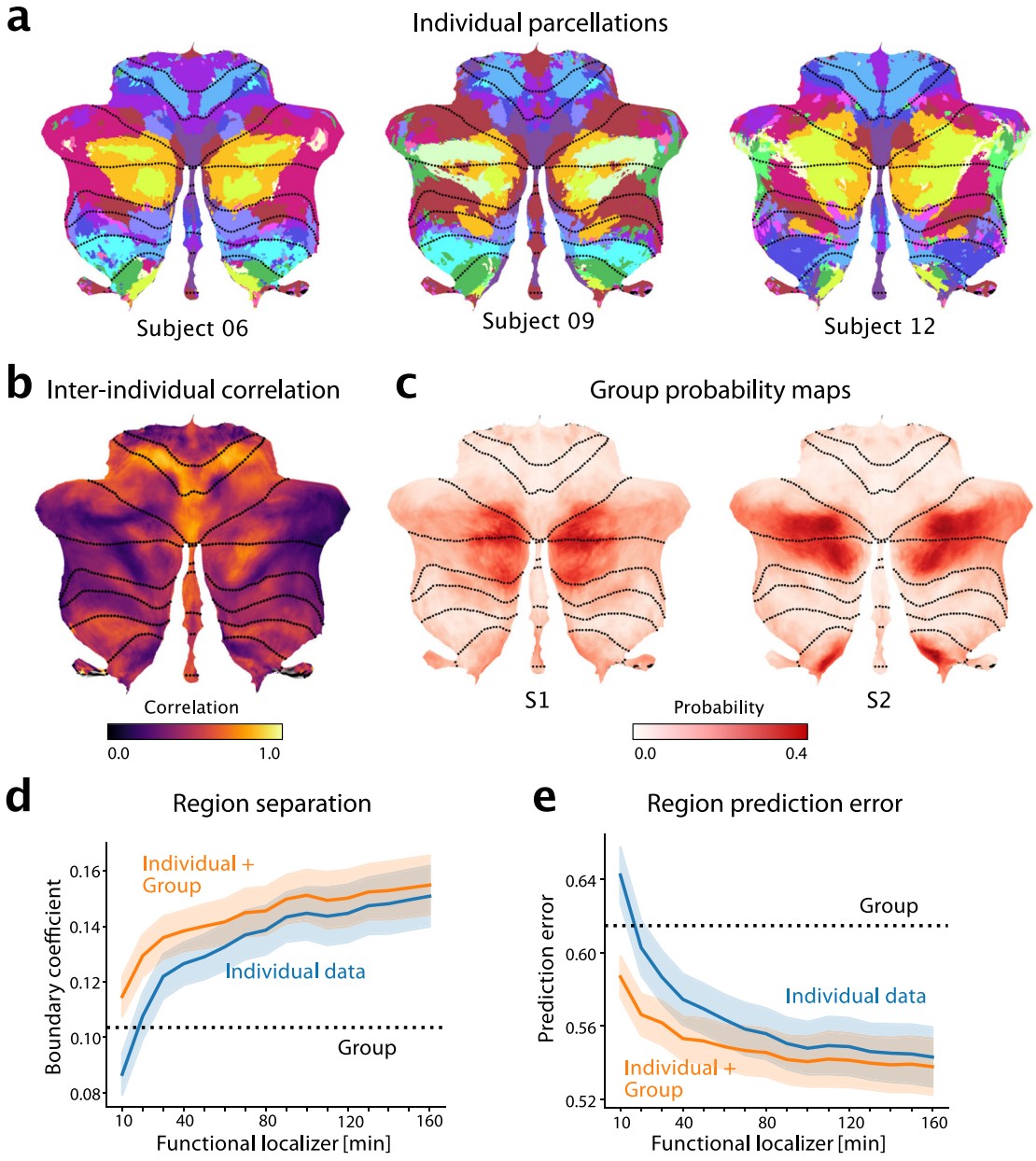

**Fig. 5 | The functional atlas improves individual precision mapping. a** Individual parcellations from three participants, using 320min of individual data. The region colors correspond to the atlas at medium granularity (32 regions). **b** Map of the average inter-subject correlations of functional profiles. Correlations are calculated between any pair of subjects in the MDTB dataset, corrected for the reliability of the data (see methods: Inter-individual variability). **c** Group probability map for regions S1 and S2 (left and right combined) show the overlap of regions. **d** DCBC evaluation (higher values indicate better performance) on individual parcellations (blue line) derived on 10–160min of individual functional localizing data, compared to group parcellation (dashed line) or the combination of group map and individual data (orange line). Shaded area indicates SE of the mean across subjects. **e** Equivalent analysis using prediction error (see methods, lower is better). Shaded area indicates SE of the mean across subjects for all datasets apart from MDTB-Highres ($n = 103$ for each bar). Source data are provided as a Source Data file.

(Verb generation)(Supplemental Fig. 6c, d). S5 was also activated by the action observation task, such that it functionally takes up an intermediate position between the social-linguistic-spatial and action domain. When comparing these regions to the recently described subdivision of the default network[26], S4 and S5 appear more similar to default network A (associated with remembering and scene construction), and S2-S3 to default network B (theory of mind).

**Cerebral connectivity patterns characterize distinct regions**
The cerebellum does not work in isolation—indeed, given the uniform cyto-architecture of the cerebellum, functional specialization arises from the different patterns of connectivity. We therefore characterized each cerebellar region by determining the areas of the cerebral cortex that most likely provide input to this area. To do so, we estimated an effective connectivity model, aiming to explain the data in each cerebellar voxel as a linear combination of cortical regions[27]. For the task-based dataset, we used the condition-averaged profiles, for the resting-state data, the preprocessed time-series. We fitted the models individually per subject and dataset. To validate these connectivity models, we tested them in how well they could predict the cerebellar activity patterns for each other dataset, using only the corresponding cortical activity patterns (see methods: Cortical connectivity).

The average correlation between the predicted and the observed activity patterns (Fig. 3a) were significantly higher than zero for all training / test combinations. One notable exception was the model

estimated on the Somatotopic dataset, which generally performed more poorly in predicting the other data sets. Connectivity models generally showed the highest predictive accuracy on the dataset they were trained on, even though this evaluation was cross-validated across subjects.

Averaged over all evaluation datasets (Fig. 3b), the model trained on the MDTB dataset performed best—with the other models being nearly equivalent in their performance (with the exception of Somatotopic dataset). To fuse across datasets, we simply averaged the connectivity weights across models. We found that average prediction performance was slightly better if it did not include the HCP dataset (0.396 vs. 0.394, $t_{102} = -1.51, p = 0.1349$). The final Fusion model (last bar in Fig. 3b) significantly outperformed the best individual connectivity model (MDTB, $t_{102} = -7.340, p = 5.322 \times 10^{-11}$). Taking into account the noise ceiling of this prediction given by the reliability of the cerebellar and the cortical data (see methods: Cortical connectivity), the model achieved a prediction accuracy of $R = 0.6840$, meaning that it predicted on average 47% of the explainable variance.

The weights of these connectivity models for each individual region (Supplementary Fig. 7, 8, 9) clearly showed connectivity with the expected cerebral regions in the contralateral hemisphere. For example, the left cerebellar hand region showed the highest connectivity with the hand region of the right primary motor cortex and somatosensory cortex, and vice versa for the right cerebellar hand region (Supplementary Fig. 7c).

To summarize these weight maps in terms of standard cortical networks, we averaged the weights within the 15 resting-state networks described in **du2023?** (Fig. 3c). This analysis showed the expected connectivity between M1 and visual and dorsal attention networks, between M2-M3 and the Somatomotor and premotor networks, D1-D4 to the dorsal Attention network A and control networks, and S1-S5 to language and default networks.

## Functional lateralization and boundary asymmetry

The symmetric version of our atlas forced the boundaries between parcels to be the same across hemispheres. Nonetheless, the functional profiles for the left and right parcels were estimated separately (see methods: Symmetry constraint). Therefore, hemispheric differences in functional specialization were captured by the model. To investigate these differences, we correlated the functional profiles of corresponding left and right voxels (Fig. 4c). We observed low functional correlations between left and right hand regions (M3). This was mainly caused by task sets that isolated left- vs. right-hand movements. Such task-dependence can be clearly seen in the foot motor region (M4), which appear functionally symmetric in the MDTB-Highres dataset, which included bilateral foot movements, and functionally asymmetric in the somatotopic dataset included separate left and right movement conditions (Supplemental Fig. 10).

In contrast, the multi-demand regions consistently show high functional correlations across left and right hemispheres for all datasets, even though the task sets included different executive functions and working memory tasks, using verbal and non-verbal material. While there might be some functional lateralization within this domain, our results suggest that their response profiles are largely symmetric and that it may be difficult to find strongly lateralized tasks in this functional domain. In contrast, the social-linguistic-spatial regions showed much lower functional correlations with substantial differences between left and right response profiles. Therefore, some functions are clearly lateralized in the cerebellum, reflected in different functional profiles for left and right regions.

Additionally, it is also possible that boundaries between functional regions themselves are asymmetric. We therefore estimated an asymmetric version of the atlas with the same functional profiles per region, but without the constraint on symmetry. Overall, the asymmetric atlas was similar to the symmetric atlas (Fig. 4a). However,

closer inspection revealed some key differences between the left and right hemispheric parcels of the asymmetric atlas, with the biggest difference observed among the social-linguistic-spatial and multiple-demand regions. When we compared the region size between the left and right regions in the asymmetric atlas (Supplemental Fig. 11), S3 and S4 had larger regions on the right, while S2, A2, and D1 were bigger on the left.

Finally, we calculated an index of boundary symmetry (see methods: Boundary symmetry) by correlating the parcel probabilities from the asymmetric and symmetric atlas. We found high boundary symmetry in motor and demand regions and low boundary symmetry in social-linguistic-spatial regions. Specifically, among the motor regions the oculomotor vermis M1 and the hand region M3 (Fig. 4c) showed high boundary symmetry. All demand regions showed high boundary symmetry with the exception of D2. In the social-linguistic-spatial regions, we observed generally low boundary symmetry, indicating that for these regions an asymmetric atlas may be most appropriate.

## Individual precision mapping through integration of localizer data

The fusion atlas reveals several finely inter-digitated regions that have not been well described before and that have only been localized at the single-subject level using large quantities of individual data[15]. However, with the probabilistic framework, the atlas can be used to identify these regions in individual participants even with more limited data. In this section, we will describe the approach of personalizing the atlas to individuals, i.e., using the atlas for precision mapping[10–12].

We first characterized the spatial pattern of inter-individual variability to understand where in the cerebellum individual localization would offer the greatest utility. For each voxel, we calculated the Pearson's correlation between the functional profiles of all possible pairs of subjects in the MDTB dataset (methods:Inter-Individual variability). While motor regions showed consistent functional profiles across subjects (e.g. hand regions M3 and eye regions M1 in Fig. 5b), the social-linguistic-spatial regions were more variable. Only voxels in the core of the S1 region were relatively consistent across individuals; the lateral regions, and especially the boundary to the multi-demand regions demonstrated large inter-individual variability. Consistent with the heightened inter-individual variability in the social-linguistic-spatial regions, our atlas shows considerable overlap in the group probability maps for region S1 and S2 (Fig. 5c). Hence, the study of these regions in Crus I and II and their differentiation from demand regions will benefit most from precision mapping of individuals.

For individual functional localization, a common approach is to acquire functional data from the individual to define individual regional boundaries[28–30]. However, a substantial amount of functional data is necessary for deriving a parcellation that performs convincingly better than a group map[3,8,14]. We quantified this problem here by using 10min-160min of imaging data from the first session of the MDTB data set to derive individual parcellations. We then evaluated these parcellations on how well they separated functional regions (DCBC, higher DCBC indicating better separation; Fig. 5d) and predicted the functional profiles (prediction error, lower error indicating better prediction; Fig. 5e). We found that 20 min of individual data were necessary to be just as good as our symmetric group atlas, and 40 min to significantly outperform the group map on both criteria (DCBC: $t_{23} = 2.981, p = 0.0067$, Prediction error: $t_{23} = -2.869, p = 0.0087$).

The probabilistic framework, however, allowed us to optimally combine evidence from the individual data with the probabilistic group map (see methods: Individual precision mapping). The final estimate of the model using only 20 min of functional localization data outperformed both the individual data (DCBC: $t_{23} = 11.468, p = 5.43 \times 10^{-11}$; Prediction error: $t_{23} = -9.098, p = 4.414 \times 10^{-9}$) and the group map ($t_{23} = 3.395, p = 0.0025$). The integrated estimate even improved

individual parcellations based on as much as 160 mins of data (DCBC: $t_{23} = 5.838, p = 5.989 \times 10^{-6}$, Prediction error: $t_{23} = -3.798$, $p = 9.288 \times 10^{-4}$). Thus the present atlas offers both the advantage of a consistent group map, as well as the possibility to obtain precision individualized mapping of brain organization.

## Discussion

### Summary

In this study, we developed a comprehensive functional atlas of the human cerebellum featuring several important advances: First, using a Hierarchical Bayesian Model, we integrated data across seven large task-based datasets, thereby achieving a more complete coverage. The present group atlas outperforms existing task-based[3] and resting-state[7] atlases in predicting functional boundaries across functional domains. Second, by enforcing boundary symmetry but letting functional responses vary between hemispheres, our symmetric atlas version is particularly suited to study functional lateralization in the cerebellum. Third, the atlas is hierarchically organized, allowing for a consistent description of the cerebellum at different levels of granularity. Finally, the probabilistic group atlas can be combined with a short localizer scan to improve functional precision mapping of individuals. As compared to the existing winner-take-all group atlases, this approach paves the way to a detailed analysis of small subregions in the future.

### Three-fold organization of the human cerebellum

Consistent with previous studies[7,15,31], we found overall a three-fold spatial organization of the cerebellum. For most regions, we found a primary representation located between lobule I and Crus I, a secondary representation between lobule Crus II and lobule VIIIb, and a tertiary representation in lobule IX or X. The ordering of the regions was mirrored around the horizontal fissure, such that the demand region formed a shell around the social-linguistic-spatial regions, and the action and motor regions a shell around the demand regions. While regions S2-S4 appeared on the flatmap[32] to be spatially contiguous, the volumetric view revealed 9 that these regions too have anatomically distinct primary and secondary representations, separated by the horizontal fissure. This observation exemplifies the importance of considering how regions are distributed on a fully unfolded cerebellar cortical sheet[33] instead of solely relying on the crude approximation that is offered by our flatmap visualization[32].

The group atlas also shows a third representations of cognitive regions in lobule IX. No third motor representation was found in the cerebellar hemispheres. Instead, a third representation of the motor regions in the inferior vermis has recently been described at the individual level using deep phenotyping approaches[21]. Our atlas, which included these data within its training set, now clearly shows this representation both at the group and the individual level 8.

Damage to the primary motor representations leads to more severe deficits than damage to the secondary motor representation[34]. Based on this observation, it has been speculated that there are functional differences between the three representations[31]. So far, however, a definite demonstration of distinct response profiles among the three representations has remained elusive. Two lines of evidence cast doubt on a strong functional dissociation between these representations. First, our analysis of functional regions generally grouped the three representations together, implying a significant degree of shared functional profiles across datasets. Second, tracing studies have shown that a single axon from the inferior olive can branch into multiple climbing fibers[35] and innervate different regions in non-contiguous lobules[36]. Similarly, most ponto-cerebellar mossy fibers project to multiple lobules[37]. This suggests that all three representations, despite their spatial separation, may receive very similar, or even shared, climbing fiber and mossy fiber inputs. Therefore, it is not clear whether the multiple representations of the same functional region can be

functionally distinguished. To facilitate further investigations, we provide an atlas version, in which each region is subdivided into a superior (lobule I–Crus I), inferior (Crus II–VIIIb), tertiary (lobule IX–lobule X), and vermal sections (vermis VII–vermis X). With one exception (S5), this subdivision separates the spatially non-contiguous aspect of each region.

### Functional insights

Although the spatial pattern of most regions adheres to a three-fold organization, our atlas reveals that several regions deviate from this principle, suggesting a more complex cerebellar functional organization. First, not all functional regions have all three representations, for example A3 and S5 only have an inferior representation, whereas M1 only has a superior representation (Supplemental Fig. 3a). Second, some regions with a primary and secondary representations are spatially connected in the volume (e.g., S1, Supplemental Fig. 3a). Future neuroimaging studies might reveal a parsimonious organization or more spatial complexity, as has been suggested by intensive within-individual mapping[15].

Furthermore, while our atlas confirms the well-known functional regions of the cerebellum, it also uncovers regions that have not been reported or only recently identified. We describe two previously unreported regions in lobules VIII and IX, notably A3 which is engaged during spatial simulation and S5 which activates when constructing an imagined scene or engaging in specific forms of self-projection. Furthermore, the atlas revealed 5 medial-to-lateral organized regions in Crus I and II. A similar detailed subdivision has only been achieved at the individual level using several hours of scan time[15,26]. This work showed that the default network can be divided into two parts, one that is associated with remembering and scene construction (network A), the other that is associated with mentalizing (network B). Our atlas captures this distinction, with S4 showing some correspondence with default network A, and S2 and S3 with default network B.

However, it is not clear a-priori that there should be 1:1 correspondence between the regions identified in this atlas and cerebral resting-state networks. Our atlas is based on data that is task-based and comes from the cerebellum only. It therefore offers a different and complementary approach to resting-state atlases, in which the networks are defined on the cerebrum, and the cerebellum subsequently labeled according to the best-matching network[7].

### Individual precision mapping

Studying finely inter-digitated regions is difficult when using group-level atlases. Inter-individual variability is generally high in the cerebellum[14], and our analysis (Fig. 5d) shows that the location and arrangement of the multi-demand and social-linguistic-spatial regions are especially variable across individuals. High inter-individual variability has been a long-standing finding for language regions. Despite this variability, the spatial pattern of the language network, its degree of lateralization and responsiveness are relatively stable within individuals over time[38,39]. These results stress the importance of using an individualized approach when studying cognitive regions of the cerebellum[40–42].

The classic approach to individual localization is to run a short localizer scan (often 10 min)[29], based on the assumption that these individual-level boundaries reflect the subject's organization better than boundaries defined by a group map, or through localization using resting-state network estimates[43]. However, experience suggests that substantial amount of scan data are required to predict individual functional data better than the group map. We confirm this by showing that the probabilistic group map provided by our atlas is as good as 20 min of individual data (Fig. 5d), rendering individual localization based on only 10 minutes of data suboptimal. Increasing the individual scan time[15] often is not feasible, especially in the clinical context.

Similarly to the Bayesian model proposed by Kong et al.[16], our atlas offers an alternative, by optimally integrating even limited individual data (10-20 minutes) with the probabilistic group map. This integration yields a probabilistic map of regions in the individual that is better than both group and individual map.

To apply this approach to a new subject in a new study, one needs to acquire some independent individual localization data (see below). Our framework can then be used to train a new dataset-specific emission model that characterizes—for each cerebellar region—the average group response on the tasks contained in that localizer scan. The final individual parcellations are obtained by combining the data likelihood with the probabilistic group map (see methods: Group and individual parcellations). This method enables the use of individual functional localization in studies for which the time with each individual is restricted. Even for longer localizer scans, our approach leads to significant improvement than using the individual data alone. The code and documentation for individual precision mapping is available at github.com/DiedrichsenLab/HierarchBayesParcel (https://doi.org/10.5281/zenodo.12976154).

An important consideration for a precision mapping approach remains the decision of whether to use task-based or resting-state data, and—if using the former—which localizer tasks to include. For many purposes, it seems advisable to include a set of anchor tasks able to activate each region of interest. We observed that task-based datasets that focused on a narrow functional domain resulted in precise estimates of boundaries for regions of that domain at the expense of region boundaries for other domains (Fig. 1a).

In addition to tasks that tap into the domain of interest, it is likely beneficial to include tasks that activate spatially neighboring regions. For example, when aiming to study the language regions of the cerebellum[29], adding tasks that activate the neighboring multi-demand regions may help to obtain a more precise estimate of the functional boundary between social-linguistic-spatial and multi-demand regions, which appear especially variable. The development of a principled approach to design optimal task-sets for functional localization remains an important question for future research.

Overall, functional precision mapping will likely be increasingly important in the future to study the function of smaller, more variable subregions, study brain connectivity[9,44], targeted neuromodulation[45–47], and individualized diagnostic and prognosis.

## Lateralization

The cerebellum's importance in lateralized higher-order functions, particularly language, has reignited interest in lateralization studies of the cerebellum[18]. Studies of hemispheric specialization are most easily performed using a functional atlas that has regions matched in size and location across hemispheres, while as closely as possible representing functional boundaries. Prior studies that examined hemispheric differences in cerebellar development[18] or neurochemistry[48] had to rely on anatomical parcellations, even though these are not good descriptions of functional subdivisions[3]. Our symmetric atlas addresses this gap, and we show that the symmetry constraint had only a relatively small impact on its ability to identify functional subdivisions.

## Cerebro-cerebellar connectivity

For each of the cerebellar regions, our framework also provides a cerebral connectivity pattern. We showed that a model that integrates data across diverse task-based dataset outperforms our previous model that was only trained on the MDTB dataset[27]. These patterns of cerebral connectivity not only provide an additional description of the identified regions but have two further practical applications.

First, being able to identify a cerebellar region by its cerebral pattern of connectivity allows the use of resting-state data to localize these regions in single individuals[7,15]. This enables the extension of the atlas to patient groups and young children and allows users to leverage the broadly available resting-state datasets.

Secondly, the independent identification of the cerebral regions that communicate with each cerebellar region is an important prerequisite for further studies that investigates the functional differences between cerebral and cerebellar areas within the same functional module[49]. We therefore believe that the present atlas will provide an important resource for the study of the human cerebellum going forward.

# Methods

## Datasets and data organization

We used seven task-based and one resting-state fMRI datasets (see Supplemental Table 1). All studies were approved by the respective institution's medical ethical committees or review board. Each of the first four datasets comprised a broad battery of tasks tapping into cognitive, motor, perceptual, and social functions: (1) The *Multi-Domain Task Battery* dataset (MDTB,[3]), (2) a high-resolution version of the MDTB (*High-res MDTB*; not yet published), (3) the *Nakai & Nishimoto* dataset[50], and the (4) The *Individual Brain Charting (IBC)* dataset[51,52]. We also included three further datasets to obtain a better description of the motor and executive functions: (5) the working memory (*WM*) dataset[49] which included finger movements and a forward / backwards digit span task; (6) the *Multi-Demand* dataset[25] which included a no-go, n-back, and task-switch task; and (7) the *Somatotopic* dataset[21] which probed foot, hand, glutes, and tongue movements. Finally, we used the resting-state fMRI dataset *Unrelated 100* subjects, which is made publicly available in the *Human Connectome Project (HCP)* S1200 release[53].

Sex and gender were not considered in the study design. and only sex, but not gender was recorded based on self-report. However, the final atlas sample was approximately balanced for sex (60 males and 51 females). Study demographics are reported in Supplemental Table 1.

The task-based datasets were preprocessed as described in ref. 8. For each run and condition, we estimated one contrast image, and divided it by the root-mean-square-error from the first-level GLM to obtain a normalized activation estimate for each condition. These values served as the input data for all subsequent analyses. No smoothing or group normalization was applied at this stage. For the HCP resting-state data, we used minimally preprocessed time series[54]. The preprocessing pipeline included correction for spatial distortion and head motion, registration to the structural data, cortical surface mapping, and functional artifact removal[54,55]. This resulted in 1200 time points of processed time series per imaging run per cerebellar voxel of the standard MNI152 template[56]. To obtain resting-state functional connectivity (rs-FC) fingerprints of the cerebellar voxels, we used a group Independent Component Analysis (ICA). We applied the group-ICA implemented in FSL's MELODIC[57] with automatic dimensionality estimation to the temporally concatenated functional data of all subjects, sessions and runs, and selected the top 69 signal components. We then regressed the 69 group network spatial maps into each subject's data, resulting in 69 subject-specific network time courses. The cerebellar rs-FC fingerprints were calculated as Pearson's correlations of the cerebellar voxel time series with each cortical network time course.

Using a unified code framework (available at github.com/diedrichsenlab/Functional_Fusion), the data were then extracted in two atlas spaces. For the cerebellum, we computed the non-linear morph into the Symmetric MNI152NLin2009aSym template (http://nist.mni.mcgill.ca/?p=904). The functional data were resampled to a group space of 18290 cerebellar gray-matter voxels with an isotropic resolution of 2mm. During this step, we only considered voxels within the individual cerebellar mask, taking care to exclude any signals from the directly abutting neocortical regions. For interpolation of functional signals within the cerebellum we used a Gaussian kernel of 2mm

standard deviation. For the cortical-cerebellar connectivity models, the same data were projected onto individual surfaces, which are aligned to the symmetric freesurfer32LR template[56].

## Hierarchical Bayesian parcellation framework

To integrate different datasets into a unified probabilistic parcellation atlas, we utilized a recently developed Hierarchical Bayesian Framework[for full details 8]. In short, the framework integrates different fMRI datasets, $\mathbf{Y}^{s,n}$, recorded in different sessions ($n$) from different subjects ($s$). The model assigns each of the possible brain locations in each individual to one of $K$ functional parcels, with $\mathbf{U}_{k,i}^s = 1$ indicating that the $i^{th}$ voxel is part of the $k^{th}$ parcel. The model estimates the expected value of these parcel assignments, which provides a probabilistic parcellation for that individual.

The model consists of two parts: First, a collection of dataset-specific *emission models* that specify the probability of each observed dataset given the individual brain parcellation, $p(\mathbf{Y}^{s,n}|\mathbf{U}^s)$. Here, we used a van-Mises-Fisher mixture model, in which each parcel had a mean vector $\mathbf{v}_k^n$ for each session, and a separate concentration parameter for each session $\kappa^{n, \text{Model Type 2, see 8}}$. Each emission model therefore had the parameters $\boldsymbol{\theta}_E^n = \{\mathbf{v}_1^n,...,\mathbf{v}_k^n,\kappa^n\}$.

The second component, the *arrangement model*, specifies the group probability of each brain location belonging to a specific parcel. Here we used a model that treated each voxel independently, with $p(\mathbf{U}_{k,i}^s) = \text{softmax}(\eta_{k,i})$. The $KxP$ arrangement model parameters $\boldsymbol{\theta}_A = \{\boldsymbol{\eta}_{1,1},...\}$ could therefore be estimated by averaging across all the individual probability maps. During this integration step, the concentration parameter for each dataset effectively determines the weight by which an individual contributes to the overall group map.

The parameters of the spatial arrangement models and the emission models were estimated together using an EM-algorithm. We used 5000 different random starting values to avoid local minima. For computational reasons, the initial fitting and evaluation was done using a 3mm isotropic voxel resolution—the final selected model was upsampled to 2mm and used as a starting value to refit to the higher resolution data.

## Symmetry constraint

To achieve spatially symmetric parcellations, we developed a version of the arrangement model, where parcels $1...K/2$ were restricted to the left hemisphere, and parcel $K/2+1,...,K$ to the right. The assignment of voxels to parcels was symmetric—that is if the left hemisphere voxel was assigned to parcel 1, the corresponding right hemispheric voxel was assigned to parcel $K/2+1$. As a consequence, symmetric brain locations were assigned to corresponding parcels. The mean functional profiles $\mathbf{v}_k^n$, however, were estimated separately for the left and right hemispheric parcels. This allowed us to derive a spatially symmetric parcellation of the cerebellum, while still capturing the functional specialization of each hemisphere.

To construct a corresponding asymmetric atlas, we removed the symmetry constraint, now allowing left and right-hemispheric voxels to be assigned to non-matching parcels. However, to retain the same number of regions, we retained the constraint that one half of the regions were in the left, the other half in the right hemisphere. To make the asymmetric atlas comparable to the symmetric version, we also used the fitted emission models (mean functional profiles) from the symmetric model, only refitting the arrangement model without the symmetry constraint. This resulted in an asymmetric version of the atlas in which the regions had the same functional profiles as in the symmetric version.

## Group and individual parcellations

After fitting the parameters $\{\boldsymbol{\theta}_A,\boldsymbol{\theta}_E^1,...,\boldsymbol{\theta}_E^N\}$, the model can be used to derive both a group and individual parcellation maps. The probabilistic group parcellation is based only on the arrangement model, which

directly specifies $p_{group} = p(\mathbf{U})$ for each voxel and parcel. Each individual parcellation is based on some *individual training* data, $\mathbf{Y}_s^n$. The data-only parcellation only depends on the corresponding emission model, with $p_{data,s} \propto p(\mathbf{Y}_s^n|\mathbf{U}_s)$. In contrast, the full individual parcellation integrates the probability from both emission and arrangement model $p_{indiv,s} \propto p(\mathbf{Y}_s^n|\mathbf{U}_s)p(\mathbf{U}_s)$, using Bayes rule. For visualization and evaluation, both group and individual probabilistic parcellation were transformed into hard parcellations by assigning each voxel the parcel with the highest probability.

## Individual precision mapping

Our model provides a probabilistic group map (spatial arrangement model) and a probabilistic estimate of parcel membership based on a specific individual data set (using a dataset-specific emission model). By integrating these using Bayes rule, an optimal estimate of brain organization for a new individual can be obtained[8]. For the analysis presented in Fig. 5, we used 1-16 runs of data from the first task set of the MDTB dataset as training. The individual maps were then evaluated on the second task set, which contained 8 overlapping and 9 novel tasks[3].

To apply this approach to new subjects with individual localizing data that is different from the task sets included in our atlas, the user would first estimate a new emission model from the data of all individuals in the study. This new dataset-specific emission model can be used to localize regions in new individuals, given their data.

## Single-dataset parcellations and similarity analysis of parcellations

To compare the differences between parcellations derived from different datasets, we trained the model on each dataset separately, estimating parcellation maps with 10, 20, 34, 40 and 68 regions. As an index of parcellation similarity, we calculated the adjusted Rand Index (ARI) between the winner-take-all voxel assignments of the resulting parcellations. The ARI was calculated across all 5 levels of granularity, resulting in a 5x5 matrix of ARIs for each dataset pair. Different datasets are differently reliable which could affect the similarity of two datasets. We therefore estimated the reliability of the parcellation by averaging the ARIs between different levels of granularity within each dataset, with the idea that reliable datasets should result in parcellations that are consistent across granularities. We then divided the ARI (also average across levels of granularity) between two datasets by the geometric mean of the two average within-dataset ARIs. This index served as a reliability corrected measure of correspondence between parcellations.

Statistical tests to compare the similarity of two data set pairs were performed using a paired t-test, using reliability-corrected ARIs for the unique 25 different granularity pairs as independent observations.

Finally, we used classic multi-dimensional scaling to visualize the structure of similarities between different parcellations. We calculated the first two eigenvectors of the square matrix of adjusted between-dataset similarities. The space defined by these two vectors optimally reproduces the overall similarity structure, with the dissimilarity (1-ARI) between two datasets reflected in the Euclidean distance between the two.

## DCBC evaluation

To assess how well a given parcellation can predict functional boundaries in the cerebellum, we utilized the Distance-Controlled Boundary Coefficient (DCBC)[22]. This metric compares the correlation between voxel-pairs within a parcel to the correlation between voxel-pairs across a boundary, while accounting for spatial distance. Our evaluation included both the group parcellation (DCBC group) and individual parcellations (DCBC individual) obtained from this group atlas.

Both group and individual DCBC were calculated in a cross-validated fashion, leaving out the test dataset during training of the overall model. The group DCBC was calculated by deriving a winner-take-all parcellation from the group probability map and evaluating the ability of these group-based boundaries to predict functional boundaries in each individual.

To calculate the DCBC for individual parcellations, we used a localizer-like approach for individual precision mapping (see methods: individual precision mapping): One half of the test dataset served as the localizer data. First, we estimated a dataset-specific emission model for the localizer dataset across all subjects. Then, we used the localizer data from one specific subject to estimate the individual boundaries (see methods: group and individual parcellations). Hard-parcellated individual boundaries were derived using a winner-take-all approach on the subject's resultant individual probability map. These were then tested for their ability to predict functional boundaries in the second half of the subject's data. We then reversed the role of the two halves of the test set averaged performance across the two within-subject cross-validation folds. To make the evaluation of group-based and individual-based boundaries comparable, we also calculated the group DCBC by splitting each subject's data in half and then averaging the performance across the two halves after individual DCBC calculation. A higher DCBC value indicates better performance of the parcellation.

### Prediction error evaluation

To assess the ability of a given parcellation to predict functional responses in individual held-out data, we calculated a prediction error. Using the same localizer-like approach as for the individual DCBC, we first derived the individual parcellations from one half of each dataset, and converted these to winner-take all maps. We then used the data from $N-1$ subjects of the second half to estimate the mean functional profiles ($\mathbf{v}_k$) for each region. For each voxel in the $N^{th}$ subject, we then used the profile of the assigned region as a prediction and calculated the prediction error as one minus the cosine similarity of prediction and data vector. When averaging these results across voxels, we weighted each cosine error by the length of the data vector to ensure that voxels with high signal strength would influence our evaluation more than noisy voxels[8].

### Parcel similarity and clustering

To develop a hierarchically organized system of maps, we started with the symmetric map with 68 parcels (34 per hemisphere) as our base. For clustering we derived a functional similarity index between parcels. We first averaged the estimated mean response vectors for each parcel and session $\mathbf{v}_k^n$ across the left and right hemisphere, and then calculated the cosine similarity between each pair of parcels. We then took the weighted average of these cosine-similarities across sessions and datasets, with the weight of each session set to product if the dispersion parameter $\kappa^n$ and number of subjects for that session $N^n$.

We then iteratively merged the smallest parcels into the functionally most similar parcel, until all parcels had at least one voxel win the winner-take-all assignment, resulting in 32 parcels (16 per hemisphere). When merging parcels, we summed their probability maps to obtain the probability of a voxel to belong to the combined parcel. The emission models for the combined model were then refit to the data, keeping the probabilities in the arrangement model fixed. In a last step, we grouped the 32 parcels (again, based on their functional profiles) into 4 domains. The labels for each parcel then followed the organization of Domain-Region-Hemisphere-Subregion.

The colormap for our functional atlas was based on the weighted cosine similarity of the functional profiles (see above). We used classical multi-dimensional-scaling to represent these similarities in a 3-dimensional space. This arrangement was then projected into RGB space. We used 3 spatial anchor points (motor region = green, demand = red, social linguistic = yellow) to achieve a consistent color scheme across parcellations (i.e. Figure 1a). As a result, the similarity of color of different parcels can be directly interpreted as an approximation of their functional similarity.

### Functional lateralization and Boundary symmetry

To study lateralization, we assessed the symmetry of the functional profiles of left-right voxel pairs. For this, we calculated the cosine similarity of the functional profiles of each voxel pair. Functional profiles were obtained by averaging the estimated mean response vectors for each voxel in each session. The cosine similarities were then weighted by the session weight $\kappa^n$ and the number of subjects $N^n$, for session $n$.

To investigate left-right boundary symmetry in the cerebellum, an asymmetric version of the atlas was estimated (see methods: Symmetry constraint). An index of boundary symmetry was calculated as the correlation between the parcel probability vectors of the asymmetric and the symmetric atlas for each voxel, either for the group map, or for the individual parcellations. For visualization, the correlation values within all datasets, excluding the Nishimoto and IBC dataset due to the relatively low reliabilities, were averaged across individuals.

### Cerebral cortical connectivity

Connectivity models were fitted for each individual (and dataset) separately. As described in King et al.[27], we parcellated the cerebral cortex into 1876 parcels using a regular icosahedron. For task-based data we used the normalized activity estimates, for the resting-state data, the preprocessed time series (see methods: Datasets and data organization). These data were averaged across all voxels in each cerebral ROI, forming the $NxQ$ matrix $\mathbf{X}$. The same data was extracted for each cerebellar voxel in atlas space. The connectivity weights were then estimated to form the best predictive model $\mathbf{Y} = \mathbf{XW}$ using Ridge-regression. The ridge coefficient was tuned for each dataset separately to yield the best prediction performance on all the other datasets.

For evaluation, we averaged the connectivity weight across all subjects in each training dataset. For each individual in the evaluation dataset, we used the cerebral cortical activity measures and the average connectivity weights to predict the individual cerebellar activity patterns. We then calculated the cosine similarity between the predicted and observed cerebellar activity[27].

When evaluating a connectivity model on the same dataset it was trained on, we adopted a leave-one-subject out approach. For each individual, the connectivity weights were averaged across all other individuals in that dataset, and then applied to make the prediction for that single subject.

Finally, we investigated if an integration across all datasets would increase the predictive power of the connectivity model. For this we simply averaged connectivity models across all task-based datasets, always taking care to leave the particular evaluation subject out of the averaging of the connectivity weights.

### Functional profiles for the MDTB dataset

To characterize the functional profile of each cerebellar region, we calculated the mean task response of all parcels in the MDTB dataset. These functional profiles were the normalized activation estimates (see methods: Dataset and Data Organization), averaged across the individualized regions within each individual. To account for activation that can be explained by the motor aspects of each task, we used the number of movements in each condition (left hand presses, right hand presses and saccades per second) as a covariate alongside regressors that coded for each condition separately[3]. The columns of the design matrix and the average functional profiles were z-normalized across conditions. We estimated a linear model using ridge regression (L2 regularization) to arrive at a final estimate for the motor features and task-activations.

### Inter-individual variability

To quantify inter-individual variability in the cerebellum, we calculated Pearson's correlation coefficient of each voxel's response profile pairwise between all subjects within the MDTB dataset. To account for the measurement noise, we derived two independent estimates for each subject and voxel: one from the first half, the other from the second half of the data. Correlations were computed on the concatenated two profiles and the reliability was calculated by correlating the two independent estimates of the response profile within each subject. The inter-subject correlation was normalized by dividing each value by the square root of the product of the two subject's reliabilities. For purposes of visualization of each voxel's inter-individual variability, we averaged the inter-subject correlation values across subjects and divided it by the reliability averaged across subjects, obtaining a single value per voxel. These voxel values were projected to the flatmap.

### Reporting summary

Further information on research design is available in the Nature Portfolio Reporting Summary linked to this article.

## Data availability

The raw fMRI data used in this study have been deposited in the openneuro database under accession code ds002105 [https://doi.org/10.18112/openneuro.ds002105.v1.1.0] for MDTB, ds005148 [https://doi.org/10.18112/openneuro.ds005148.v1.1.0] for WM, ds002306 [https://doi.org/10.18112/openneuro.ds002306.v1.1.0] for Nishimoto and ds000244 [https://doi.org/10.18112/openneuro.ds000244.v1.0.0] for IBC. For the HCP dataset, raw and preprocessed data is available at https://www.humanconnectome.org/study/hcp-young-adult/data-releases. The MDTB-Highres and Somatotopic dataset have not yet been openly released. The fMRI-derived data generated in this study are provided in the Supplementary Information and the Source Data file (https://github.com/DiedrichsenLab/ProbabilisticParcellation/blob/main/data/source_data.xlsx).

## Code availability

For a practical example on how to generate individual cerebellar parcellations using a new dataset, see https://hierarchbaysparcel.readthedocs.io/en/latest/indiv_parcel.html The code for the hierarchical Bayesian parcellation framework is available at https://github.com/DiedrichsenLab/HierarchBayesParcel. The organization, file system, and code for managing the diverse set of datasets is available at https://github.com/DiedrichsenLab/Functional_Fusion. The code for building the atlas and generating the results and figures in this paper is publicly available as the GitHub repository https://github.com/DiedrichsenLab/ProbabilisticParcellation. The code for connectivity modeling is available at https://github.com/DiedrichsenLab/cortico_cereb_connectivity. For a tutorial on how to apply the connectivity model to new data to make predictions, see https://github.com/DiedrichsenLab/cortico_cereb_connectivity/blob/main/notebooks/0.Application_example.ipynb.

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

## Acknowledgements

This study was supported by a Discovery Grant from the Natural Sciences and Engineering Research Council of Canada (NSERC, RGPIN-2016-04890), and a project grant from the Canadian Institutes of Health Research (CIHR, PJT-507612), both to J.D. Additional funding came from the Canada First Research Excellence Fund (BrainsCAN) to Western University and a National Institute of Mental Health (MH124004) to R.L.B.. We would like to acknowledge Suzanne Witt for help in collecting and preprocessing the MDTB-Highres dataset. Special thanks to M. Assem, and J. Duncan for sharing their datasets before the official public release.

## Author contributions

C.N. and J.D. conceived the study, R.B. and J.D. guided the study design, C.N., L.S., A.P., and N.S. collected the data, J.D. guided the analysis, D.Z. provided data analysis tools, C.N., L.S., A.P., and J.D. performed the analysis, C.N. and J.D. wrote the manuscript.

## Competing interests

The authors declare no competing interests.
