## [Peer Review File · Nature Communications]

A hierarchical atlas of the human cerebellum for functional precision mappingReviewer #1 (Remarks to the Author):

The authors combine several large datasets to create a novel fusion atlas. The fusion atlas allows for better detection of boundaries, increased power to conduct precision neuroimaging - even with limited individual subject data, and seems to even identify some novel areas. The atlas is an addition to the field which struggles to identify and characterize functional regions in the cerebellum. To understand the contribution of the work, the authors should more clearly delineate this atlas / current study from their prior work.

Major issues

It's slightly unclear to me what the contribution is beyond the other atlas from this group – the MDTB atlas. Many of the tasks represented in this current atlas are taken from that study. It seems like the biggest contribution to the increased power of the fusion dataset comes from MDTB (as HCP is not included in several metrics, somatotopic has low predictive power, etc.) and in fact the mean evoked responses used to characterize subregions come from the MDTB atlas. Are some of the newer subregions that are identified (S5 for example) only identifiable in the fusion atlas – or could these be found in MDTB as well? Could the authors more precisely discuss what this atlas provides beyond the MDTB?

Fig 1 - The somatotopic dataset is capturing something in posterior cerebellar regions that are socio-linguistic in the MDTB dataset. If no cognitive tasks were conducted in that dataset, the authors should speak to what the interpretation of those parcels are. In this case, using a similar color scheme is slightly confusing as people may assume they correspond to tasks. The authors may want to consider a key with parcel # (if the color refers to something uniform across parcellations).

Is it obvious that the fused dataset will perform better re boundaries given that there are more non-overlapping tasks when using more datasets?

What are the downsides of using the winner-takes all approach? Prior studies (LeBel 2021 J Neurosci and King et al., 2023 Elife) suggest greater integration or convergence of cortical inputs in certain region of the cerebellum. How are these realities reflected in a winner-take-all approach?

Minor issues

The introduction might be better organized and a little more readable by first discussing the point of parcellation, then covering the shortcomings of existing methods (with a focus on the cerebellum), and finally previewing the results regarding the new atlas. As currently organized, the reader goes back and forth between old findings/shortcomings and new results which is a little bit confusing.

Line 22: parcel should be “parcellate”

Line 60: The authors mention the “state” of the brain across tasks and how that may affect the resulting maps, but do not mention individual differences in traits/demographics that may also

heavily shape the resulting maps. They may want to discuss this point here as well.

Line 68: The authors state “On the other hand, the ability to distinguish regions among the motor and cognitive regions differed between datasets”. Do they perhaps mean “...distinguish specific regions within motor and cognitive regions”

Given that Nature Comms places methods at the end, the authors may want to preview the names of the datasets and the overall approach at the beginning of the results (or the end of the intro).

Line 92: missing “to” in “... more similar MDTB”

Line 232: “To determine these regions” should maybe read “to characterize these regions”

Figures (such as 5 B and C) – readers would benefit from the unit added to the color bar (correlation coefficient or t-value or # of subjects).

Reviewer #2 (Remarks to the Author):

The authors present a functional cerebellar atlas based on multiple data-sets, including hierarchical levels of organization reflected increasing levels of precision, symmetric and asymmetric versions, and means for integrating the resulting group atlas with a localizer scan (per subject) to produce parcellations that reflect the individual’s unique functional boundaries. The thoughtful and methodological approach is unique and commendable, and the end products will be useful to a wide range of researchers including those who already investigate the cerebellum and those who should. In general the work is excellent and provides both a significant methodological addition to the literature (with potential applications in other domains) and a step change in our ability to accurately map the functional boundaries of the human cerebellum - which is fundamental to a more complete and holistic understanding of brain function and behaviour. Overall, this is an impressive work and a valuable contribution to the field.

General comments:

1. There are generally very few issues that would need to be addressed to be suitable for publication; however, a general area of improvement would be to include additional discussion on the value of the resulting atlases over existing ones and whether/how the improvements afforded by the proposed approach would meaningfully translate to researchers who may implement them in their studies.

2. Similar to 1., while the delineation and focus on methods is excellent, it may be helpful to provide additional context on how to results expand our understanding of cerebellar organization, and what the authors see as the downstream implications of this work.

3. The finding that there is greater variability in the functional boundaries of the cerebellar regions more involved in cognitive functions is very interesting and should be highlighted and further discussed.

More specific points:

l485: Was smoothing performed only within the grey matter mask or did it include non-GM voxels as well? Please clarify. If not only within the mask, please comment on how this could potentially affect the results given the tight packing and cerebellar folding. Given the desired specificity of the results, it is important to consider how volumetric smoothing may 1) potentially blur results across adjacent (but non-contiguous) regions/lobules and 2) lead to differences in the inclusion of white matter signal that is dependent on whether or not the region is in a sulcus or gyrus (white matter partial voluming will likely be more of an issue at the "base" of lobules (i.e., adjacent to the core of the middle cerebellar peduncle) and decrease towards the gyral crowns - though this is likely to differ according to the size of the lobules and strongly interact with the original resolution.

Connected to the above point, there should be some presentation/discussion to indicate the limitations of fMRI in the cerebellum and how this may potentially affect the parcellations.

l142: Greater granularity appears to be better overall - if this is the case, why were higher levels of granularity not tested and/or what is the tradeoff that was optimized to choose this level.

l231: This section may benefit from some additional context in the beginning as to why the author included this analysis and what it is contributing to the overall picture.

l296 section: The results here are very compelling, but it may have been interesting to include an analysis combining individual resting state data with the probabilistic group map. Would there be any added value over the group atlas? Does it/would it be expected to muddy the somatomotor regions too much as described earlier? Given the ubiquity of resting state acquisitions, many researchers are likely to be highly interested in this possibility.

l421: The point about potential "anchor tasks" in the discussion is very interesting and potentially of very high importance, additional elaboration here would be useful.

Figure 3c: Please comment on the magnitude of the correlations identified between the regions and cerebral cortical networks (which is echoed in the findings in the supplementary materials) - was this expected and how does it compare to previous work?

Given the challenge of displaying results on the cerebellum in a meaningful way (both the flatmaps and the volumetric representations each having their own limitations), it would potentially be useful to include, perhaps in the supplementary materials, the atlas rendered in a 3D view of the cerebellar surface from different perspectives.

Minor points:

l15-16: This statement could benefit from added precision. There is substantial work and a reasonable understanding on the contribution of the cerebellum specifically to motor behaviours, it is more its contribution to cognitive/social behaviours that has been elusive.

l120-122: It is not entirely clear what is meant by constraining the atlas to have "spatially symmetric regions across left and right, while allowing different functional profiles". Please provide some clarification.

l190: "Executive control" can be a bit of nebulous concept with many possible functions under that umbrella (e.g. inhibition, cognitive flexibility, self-monitoring, etc.) A few words on what this means here would help to clarify.

l377: "It is presently ..." - as written this statement is not entirely clear.

l387: As "Social function" is estimated by very particular tasks this inference might be a little bit of an over generalization. Considering that "social function" is a massive concept, it may be better to be more precise here.

l396-399: This is fascinating result, but it would be interesting to provide some additional perspectives as to why this might be the case. Is this heterogeneity also seen at the cortical level? Is the author confident that this organization can be so different across individuals, or could it be somehow related to methodological reasons? Also, how confident should we be that the results in an individual are stable over time?, is it possible that instability here may contribute to the observed heterogeneity?

l410: Here the author mentions that 10 minutes of individual data combined with the group map is better than both the group and individual maps, but in the results (line 323) seem to suggest that 20 minutes is necessary to outperform these. Please clarify.

l421-431: Can the author please provide a brief hypothetical example of what the task battery might look like for a 10-minute localizer?

l489: "different dataset" -> datasets

Figure 1b: missing labels for axes

Figure 3a/c: missing label on colorbar

Reviewer #2 (Remarks on code availability):

I have not had a chance to review the code.

Reviewer #3 (Remarks to the Author):

This manuscript describes a new atlas to the already existing body of cerebellar atlases, with the specific advantage of being multi-level and optionally symmetric. The authors also outline an idea of warping individual data based on functional responses, which is interesting and potentially very valuable for cerebellar research. The methodology, data analysis and conclusions are all sound. A significant amount of work is crammed into this manuscript, and as a result the level of detail is not sufficient to really reproduce the presented results without additional information (which is likely to be provided on the authors' website in future).

I do wonder how this function-driven alignment fits with the underlying anatomy, as the space of the atlas is still defined by the structural boundaries of the cerebellum. As the authors also

point out, the lobular structure does not really provide an appropriate scaffold for functional mapping of the cerebellum, so it is a little confusing that the authors go back to this lobule-defined space to present the atlas in.

As the atlas is not dramatically different from the existing MDTB atlas, new functional insights are somewhat limited, this is more the presentation of a new tool for cerebellar imaging.

The figures are beautiful, but difficult to read, at least without digging through the text and looking up some of the references, as legends and axis labels are missing in multiple places.

Specific points:

Introduction

For readability, follow line and order of reasoning of the abstract, or vice-versa.

Results

Generally, the reader is not guided very much through the figures. The points below are just questions that occurred to me while reading. I suspect that many are answered in ref (8), so I put these points here mostly to help make the current manuscript independently readable.

Figure 1 starts with a map labelled 'MDTB'. However, this is not the atlas that the same lab has been sharing as the one generated from the MDTB database

(<https://www.diedrichsenlab.org/imaging/mdtb.htm>). Is this a modified version?

Figure 1a is lacking a legend. Do same colours in the different maps refer to the same networks?

The sentence 'A strong boundary between ...' (ln 67) should be supported by some visual guidance in the figure. How can the reader see that 'the somatotopic dataset did not delineate cognitive regions in lobule VII well' ?

Line 73: consistent boundaries – consistent with what?

Figure 1b is missing labels on the x and y axis. What information is provided here?

Line 92: were significantly more similar to the MDTB

Is the figure reference on line 128 correct? The text jumps unexpectedly from Figure 1 to 4 here.

Is the line of missing voxels in Figure 4e the symmetry axis? I would have expected this to be straight.

Line 157: its

The term 'action' as opposed to 'motor' is not completely intuitive, at least not to me.

Figure 3 misses titles for the subpanels. As 'Fusion' is included with an error bar in panel b, it is not clear why it is left out of panel a. Why was the MTBD-highres not included here?

Line 280: Additionally, it is

Figure 5 – panel (a) lacks a legend.

Does it matter which 20 or 10 min of individual data are used to estimate the individuals' map?

Presumably some combinations of tasks are more useful than others. A reference to the methods or short description of a suggested localiser task set would be useful for replication.

The individual data used here is drawn from the group of participants the atlas was based on.

The size of the databases is perhaps sufficient to avoid circularity, but would the mapping also work for new participants, e.g. scanned on a different scanner (location/vendor/field strength)?

Line 405: "predict individual functional data better than the group map." (there are similar sentences elsewhere in the text). What does the term 'better' reflect here? Higher DCBC values?

Line 557: to compare the the similarity

Line 587: grammar? Please check

Response to Reviewers

We thank the reviewers and editor for their positive assessment and their helpful and constructive comments. We have carefully considered all the issues raised and made changes to the manuscript accordingly. We hope that they will agree that the manuscript has been considerably strengthened as a result.

We have responded to each of the points made individually below, including any relevant excerpts from the manuscript in each case. All changes to the manuscript have been highlighted in the revised document.

Reviewer 1

The authors combine several large datasets to create a novel fusion atlas. The fusion atlas allows for better detection of boundaries, increased power to conduct precision neuroimaging - even with limited individual subject data, and seems to even identify some novel areas. The atlas is an addition to the field which struggles to identify and characterize functional regions in the cerebellum. To understand the contribution of the work, the authors should more clearly delineate this atlas / current study from their prior work.

We would like to thank the reviewer for their constructive review of our manuscript. Their comments have substantially improved the quality of the manuscript, particularly where the advances this atlas makes are highlighted.

Major issues

- 1. It's slightly unclear to me what the contribution is beyond the other atlas from this group – the MDTB atlas. Many of the tasks represented in this current atlas are taken from that study. It seems like the biggest contribution to the increased power of the fusion dataset comes from MDTB (as HCP is not included in several metrics, somatotopic has low predictive power, etc.) and in fact the mean evoked responses used to characterize subregions come from the MDTB atlas. Are some of the newer subregions that are identified (S5 for example) only identifiable in the fusion atlas – or could these be found in MDTB as well? Could the authors more precisely discuss what this atlas provides beyond the MDTB?*

We regret that the specific contributions of this atlas beyond the MDTB atlas have not become clear and thank the reviewer for the opportunity to clarify these. While the MDTB map was based on a single dataset of 24 subjects and 62 task conditions, the fusion atlas was derived from 7 fMRI datasets including 111 subjects and 417 task conditions. Hence, while the MDTB dataset influenced the fusion atlas and appears similar on the flat map, there are substantial differences between the two, owing to the additional information captured by the 87 subjects and 355 tasks of the other 6 datasets.

To better understand the contributions from datasets other than the MDTB, we calculated the similarity of the atlas at medium granularity (32 asymmetric regions) with the parcellations resulting from the individual datasets (Somatotopic, Demand, WMFS, IBC, Nishimoto, MDTB-Highres) and the existing MDTB parcellation with 7, 10, and 17 regions. To quantify this similarity,

we used the adjusted Rand Index (ARI) restricted to the voxel pairs with at least one of the voxels located in the area of interest. We reasoned that if the delineation of a region was influenced by a particular dataset, it should appear most similar to that dataset's parcellation. To summarize the results, we assigned each parcel of the fusion atlas to the dataset it had the highest ARI with.

The large majority of regions appear most similar to regions from datasets other than the MDTB. In the motor regions, right M3 as well as bilateral M1, M2 and M4 have a most similar shape as in the Nishimoto, IBC and Somatotopic parcellation. The foot regions M4 were expectedly most similar to the dataset with extensive mapping of foot movements, the Somatotopic dataset. Indeed, this is a region that was not well delineated in the original MDTB atlas at all.

All Demand regions (D1-D4) show the highest similarity with the subdivision in the WMFS dataset, which probes working memory and executive functions. Finally, while left S1 appears highly similar to the existing MDTB parcellation, the majority of social-linguistic-spatial regions (S1-S5) appear to be influenced by a variety of datasets other than the MDTB, including Demand, MDTB-Highres dataset, WMFS, IBC, and even the Somatotopic dataset.

The resulting similarity pattern shows that most of the regions we describe in this new atlas cannot be found in this form in the existing MDTB parcellation. In particular, the previously undescribed region S5, cannot be found in the MDTB parcellation. S5, which is likely involved in scene construction, appears similar to parcellations based on the Demand and MDTB-Highres dataset. Both datasets include a large number of task conditions where the participant is required to recall or reconstruct an image in their mind's eye.

More importantly, we would like to clarify the added capabilities of this atlas, which go beyond what the MDTB and other existing functional atlases have been able to provide.

I. Probabilistic atlas + Individual parcellations

Existing functional atlases, including the MDTB, provide only winner-take-all maps. These hard parcellations contain no information about the certainty with which a voxel can be assigned to a region across the population. For areas with high inter-individual variability, this poses a major problem: A voxel that belongs to a language area in 95% of people is not differentiated from a voxel that belong to the language region in 52% of people. This prevents researchers from defining ROIs that are as conservative or inclusive as is appropriate for their question. It also precludes hard parcellations from being used in a

Bayesian precision mapping approach for individual brain parcellations, as this requires a group prior that contains information about the certainty with which a voxel can be assigned to a particular region. The fusion atlas solves this problem, by providing for each region a **probabilistic** group map that is suitable for personalization using our Bayesian Hierarchical model or other probabilistic approaches. We show that combining the fusion atlas with a short localizer scan (10-20 min) allows us to derive individual functional regions equivalent to 30 minutes of scan data.

II. Hierarchy

While the MDTB parcellation exists in 3 versions at different granularities (7, 10, and 17 regions), these versions are independent of each other. That is, the region boundaries of the MDTB 17 parcellation fall into different places than the boundaries of the 7 and 10 parcellation. Hence, regions of a finer granularity cannot be summed together to obtain corresponding regions of a coarser granularity, making it difficult to relate the different levels to each other. The same is true for resting-state parcellations available with 7, 10, and 17 regions. The fusion atlas provides a **nested hierarchy** of regions. The atlas boundaries fall into 4 domains (motor, action, demand & social-linguistic-spatial), which can be divided into 32 regions and further into 68 subregions. We, therefore, provide for the first time a functional atlas with which researchers can describe the cerebellum at multiple levels of granularity that directly relate to each other.

III. Symmetry

The MDTB parcellation divides the cerebellar cortex up very differently in the left hemisphere compared to the right. This is a problem for applications that require a consistent description of the left and right cerebellar hemisphere, for example lateralization studies or studies of the developmental trajectories of left and right hemispheres. For a symmetric parcellation, most researchers therefore turn to lobules, sacrificing the functional validity and interpretability of their regions of interest. We therefore provide a **symmetric version** of the fusion atlas, by constraining the boundaries to be the same across hemispheres, with minor reductions in predictive ability and without imposing constraints on the functional responses in the regions themselves, as these can vary between left and right. Using this symmetric atlas, functional asymmetries in the cerebellum can now be studied controlling for region size and location. Since the release of this new atlas, other groups have already successfully used the symmetric atlas to study cerebellar asymmetries and found it to outperform the lobular atlas in describing structural asymmetry (Wang et al., 2024).

IV. Connectivity

The MDTB parcellation contains no information about how each region connects to the neocortex, whereas we provide connectivity weight maps for each region of the fusion atlas. This is useful for researchers wanting to select a cerebellar region of interest that is connected to a specific region in the neocortex. It also allows the localization of individual cerebellar regions by their neocortical resting-state connectivity fingerprint, opening the opportunity for individual localization in datasets where task data is not available, including patient populations and young children.

We have clarified the main advance of the novel atlas over existing parcellations in abstract, introduction and discussion (summary).

2. Fig 1 - The somatotopic dataset is capturing something in posterior cerebellar regions that are socio-linguistic in the MDTB dataset. If no cognitive tasks were conducted in that dataset, the authors should speak to what the interpretation of those parcels are. In this case, using a similar color scheme is slightly confusing as people may assume they correspond to tasks. The authors may want to consider a key with parcel # (if the color refers to something uniform across parcellations).

We thank the reviewer for raising this interesting point. It is worth noting that, while the somatotopic dataset consists of tasks mapping individual body movements, all of these are modelled against rest, during which participants can engage in a variety of cognitive functions. The somatotopic dataset can therefore plausibly capture some information that is not movement-related, as this can occur during rest. The majority of information should, however, be concentrated in the motor areas for the somatotopic dataset. Nevertheless, due to the nature of winner-take-all assignments, it can appear as if the somatotopic dataset delineates functional regions in the posterior cerebellum, even when its probabilistic assignments contain little information here.

To visualize the information captured by the different datasets, we calculated the entropy ($-\sum(p \cdot \log(p))$) of the probabilistic parcellations of the single datasets for each voxel and plotted it on the flatmap. In high entropy areas the probabilistic atlas has high uncertainty between different

parcels – in low entropy (dark) the assignment to a single parcel is relatively certain. Here, low entropy indicates voxels that are very clearly assigned to some regions over others, suggesting that this dataset is information-rich in this region. As expected, entropy is low for the somatotopic dataset in the motor regions, particularly the foot regions. However, we also find the somatotopic dataset contains some information in core social-linguistic-spatial regions.

We regret the confusion regarding the colour scheme of parcellations in figure 1a and have added a brief explanation regarding the colour scheme to the legend of figure 1a:

“Figure 1. Building a functional atlas of the cerebellum across datasets. a. Parcellations (K=68) derived from each single dataset. The probabilistic parcellation is shown as a winner-take-all projection onto a flattened representation of the cerebellum. **Functionally similar regions are colored similarly within a parcellation (see methods: parcel similarity). Across parcellations we rotated the color assignment using three spatial anchor points in multi-demand, motor, and social linguistic regions. [...]**”

As the assignment of numbers to parcels is not meaningful in figure 1a (and there is no 1:1 correspondence of parcels across maps), we have refrained from adding a key with parcel numbers to this figure.

3. Is it obvious that the fused dataset will perform better re boundaries given that there are more non-overlapping tasks when using more datasets?

While it may appear that adding more data to a functional parcellation will always increase its ability to predict functional boundaries, this is not the case. If you simply weight all data the same, then adding data of low quality could potentially decrease prediction performance. For example, the Nishimoto and IBC dataset, despite including the largest number of task conditions, showed lowest reliabilities and captured the least information (see entropy plots in response to previous comment). Blindly integrating both datasets with the MDTB dataset, which includes fewer task conditions but has higher reliability, will substantially reduce the ability of the resulting parcellation to predict task boundaries. To prevent this from happening, our Bayesian Hierarchical framework estimates the reliability (i.e. measurement variance) of each dataset as part of the emission model and weighs the different datasets accordingly.

However, we also provide some evidence that different functional domains emphasize different functional boundaries (Figure 1b). That is, even if the data is integrated perfectly using Bayes Rule, it is possible that adding a specific dataset may decrease the predictive accuracy for other datasets.

Note also that the evaluation (Figure 1e) of each atlas map was performed on independent data, such that the exact same task conditions were never both in training and test set.

4. What are the downsides of using the winner-takes all approach? Prior studies (LeBel 2021 J Neurosci and King et al., 2023 Elife) suggest greater integration or convergence of cortical inputs in certain region of the cerebellum. How are these realities reflected in a winner-take-all approach?

We are not exactly sure how to interpret your question.

1. On the group-level, a winner-take-all parcellation ignores the substantial inter-subject variability. Our probabilistic atlas provides a measure of the variability of a specific functional boundary across the population. It can therefore be used in a Bayesian framework to integrate individual data to obtain a posterior probability. We hope that this point is relatively clearly expressed in the paper (line 32-47; line 347-).
2. On the individual level, we do assume that each voxel can be assigned to a single specific parcel. However, if a voxel shows a mixture between the activity profiles from two neighboring parcels, the probabilistic individual parcellation will assign some probability weight to both parcels. Note, however, that when visualizing the parcellation, and when evaluating the parcellation using the boundary coefficient, we need to transform the probabilistic parcellation into a hard parcellation. When using the prediction error (Figure 5e), the probabilistic (mixed) assignment is taken into account. To strike an appropriate balance between the easy visualization offered by the winner-take-all maps and the details captured by the probabilistic group maps, we chose to include both the group winner-take-all map (figure 2a-c) and the group probability maps of two exemplary regions (S1 and S2 in figure 5c) into the main text and added the probability maps of all other regions to the supplementary material.
3. Finally, classical connectivity models between neocortex and cerebellum assume a 1:1 region-to-region connectivity at the group level (e.g. Buckner et al. 2011). In the paper by King et al. (2023), we use individualized approach that also allows for the convergence of multiple neocortical regions onto a single cerebellar voxel. The connectivity models presented in this paper follow exactly the same approach, thereby allowing for a mixture of cortical signals to predict the activity in each cerebellar voxel.

Minor issues

5. *The introduction might be better organized and a little more readable by first discussing the point of parcellation, then covering the shortcomings of existing methods (with a focus on the cerebellum), and finally previewing the results regarding the new atlas. As currently organized, the reader goes back and forth between old findings/shortcomings and new results which is a little bit confusing.*

Thank you for your suggestion. In earlier versions we had indeed tried a sequence similar to those you suggest – but found that we then had to jump back and forth between different topics (fusion, individual parcellation, symmetry, hierarchy), first outlining all current limitation and later returning to the same topics with the solution suggested here. Overall, therefore we preferred the solution to the introduction suggested here.

6. *Line 22: parcel should be “parcellate”*

Corrected – thanks!

7. *Line 60: The authors mention the “state” of the brain across tasks and how that may affect the resulting maps, but do not mention individual differences in traits/demographics that may also heavily shape the resulting maps. They may want to discuss this point here as well.*

The participants in the fMRI datasets included into the atlas were largely young, healthy college students. The HCP, MDTB, MDTB-Highres, WMFS, Demand and IBC dataset largely included Caucasian participants from North America or Europe (UK and France) and the Nishimoto dataset

Asian participants from Japan. The reviewer raises an interesting point, that functional regions of the cerebellum might be affected by traits or demographic features such as age, sex or disease status. The atlas easily lends itself to study the development of functional regions across the lifespan and their associations sex, disease and traits by integrating it with the vast existing datasets from patient populations and participants with heterogeneous traits and demographics. While these investigations are beyond the scope of this paper, we hope that the open release and extensive documentation of the atlas and associated tools will spur new investigations into these questions.

8. *Line 68: The authors state “On the other hand, the ability to distinguish regions among the motor and cognitive regions differed between datasets”. Do they perhaps mean “...distinguish specific regions within motor and cognitive regions”*

Corrected.

9. *Given that Nature Comms places methods at the end, the authors may want to preview the names of the datasets and the overall approach at the beginning of the results (or the end of the intro).*

Thank you for this helpful suggestion. We have now added a short overview over datasets and the model in the beginning of the first two results sections.

“We trained our probabilistic parcellation model on ~~each~~ seven task-based and one resting-state datasets (Supplemental Table. 1) in isolation and then compared the resultant parcellations (Fig. 1a).”

“Our Hierarchical Bayesian Parcellation framework allows for data fusion by modelling each dataset separately and then combines them iteratively into a common group atlas. In this process, each dataset is weighted by a measure of its reliability (see methods, Hierarchical Bayesian parcellation framework).”

10. *Line 92: missing “to” in “... more similar MDTB”*

Corrected.

11. *Line 232: “To determine these regions” should maybe read “to characterize these regions”*

This sentence was confusingly phrased because of the use of ‘regions’ for the cerebellum and the cerebrum. We improved the phrasing of the sentences in Line 232 to clarify:

“Each cerebellar region can also be characterized by the areas in the cerebral cortex that it is most functionally correlated with. To determine the neocortical areas that were functionally connected to each cerebellar region we estimated an effective connectivity model, aiming to explain the data in each cerebellar voxel as a linear combination of cortical regions.”

12. *Figures (such as 5 B and C) – readers would benefit from the unit added to the color bar (correlation coefficient or t-value or # of subjects).*

Thank you – corrected!

Reviewer 2

The authors present a functional cerebellar atlas based on multiple data sets, including hierarchical levels of organization reflected in increasing levels of precision, symmetric and asymmetric versions, and means for integrating the resulting group atlas with a localizer scan (per subject) to produce parcellations that reflect the individual's unique functional boundaries. The thoughtful and methodological approach is unique and commendable, and the end products will be useful to a wide range of researchers including those who already investigate the cerebellum and those who should. In general, the work is excellent and provides both a significant methodological addition to the literature (with potential applications in other domains) and a step change in our ability to accurately map the functional boundaries of the human cerebellum - which is fundamental to a more complete and holistic understanding of brain function and behaviour. Overall, this is an impressive work and a valuable contribution to the field.

We thank the reviewer for their positive evaluation and their suggestions to improve the work.

General comments:

1. *There are generally very few issues that would need to be addressed to be suitable for publication; however, a general area of improvement would be to include additional discussion on the value of the resulting atlases over existing ones and whether/how the improvements afforded by the proposed approach would meaningfully translate to researchers who may implement them in their studies.*

We have included a more extensive and detailed discussion of the advances provided by this atlas over existing ones into the response to comment 1 of Reviewer 1.

2. *Similar to 1., while the delineation and focus on methods is excellent, it may be helpful to provide additional context on how the results expand our understanding of cerebellar organization, and what the authors see as the downstream implications of this work.*

Thank you for these suggestions – we have added a short paragraph at the end of the individual precision mapping section in the discussion. While we agree that the methods have multiple important application areas, a full discussion of the downstream implications of this work is not possible within the word limits.

3. *The finding that there is greater variability in the functional boundaries of the cerebellar regions more involved in cognitive functions is very interesting and should be highlighted and further discussed.*

We highlight these findings in the Individual precision mapping sections of the results (3rd paragraph) and in the discussion (1st paragraph). We have tried to extend our discussion and improve its clarity within the space constraints of the journal.

“High inter-individual variability has been a long-standing finding for language regions. Despite this variability, the spatial pattern of the language network, its degree of lateralization and

responsiveness are relatively stable within individuals over time (Mahowald et al., 2016; Fedorenko et al., 2024).”

More specific points:

- 4. l485: Was smoothing performed only within the grey matter mask or did it include non-GM voxels as well? Please clarify. If not only within the mask, please comment on how this could potentially affect the results given the tight packing and cerebellar folding. Given the desired specificity of the results, it is important to consider how volumetric smoothing may 1) potentially blur results across adjacent (but non-contiguous) regions/lobules and 2) lead to differences in the inclusion of white matter signal that is dependent on whether or not the region is in a sulcus or gyrus (white matter partial volume effects will likely be more of an issue at the "base" of lobules (i.e., adjacent to the core of the middle cerebellar peduncle) and decrease towards the gyral crowns - though this is likely to differ according to the size of the lobules and strongly interact with the original resolution.*

Minimal smoothing (Gaussian kernel of 2mm SD) was applied during the resampling of the functional data from native space to the Symmetric MNI152NLin2009aSym template. During this step, we considered only voxels located in the individual cerebellar grey matter mask, taken care that voxels extending into the directly abutting neocortical regions were not used. This is important to avoid the strong visually-driven neocortical signals from biasing the activity profiles in the superior cerebellum. These details are now provided more clearly on line 496-504.

The intrinsic spatial resolution of the different dataset varied between 1.5 and 3mm. Even at the finest resolution, however, single voxels will average signal from neighbouring folia and lobular subdivisions. Given the extremely dense folding, even voxels located in the center of the gray-matter mask, will have some partial volume effects between gray matter, white-matter and CSF. Indeed, voxel sizes of ~0.2mm would be necessary to start to cleanly separate these tissue compartments (Serenó et al. 2020).

- 5. Connected to the above point, there should be some presentation/discussion to indicate the limitations of fMRI in the cerebellum and how this may potentially affect the parcellations.*

While fMRI can provide direct insights into functional responses in the cerebellum, their diversity and spatial distribution, the technique has several limitations.

First, the cerebellar BOLD signal that fMRI measures reflects primarily input to the cerebellum. This has been shown by vasodilation studies where stimulation of the fibres transporting the input to the cerebellum – mossy fibres, parallel fibres and climbing fibres – resulted in robust increases in cerebellar blood flow, but not increases in Purkinje cell firing. While these studies suggest that the main determinant of cerebellar BOLD signal is the input into the cerebellar cortex rather than its output, this does not preclude fMRI from providing a meaningful division of the cerebellum into functional regions. Indeed, it is the climbing fibre input that is thought to determine the function of a cerebellar module. Hence, a parcellation based on fMRI responses will still accurately delineate cerebellar functional regions.

Second, fMRI suffers from a low signal-to-noise ratio in the cerebellum. This is because the cerebellum lies close to the brain stem on the anterior side, resulting in high physiological noise, and close to air-tissue cavities at the lateral hemispheres which induce strong field

inhomogeneities, resulting in signal dropout (Brooks et al., 2013). Additionally, the MR acquisition coil is often less sensitive to cerebellar signal since the cerebellum lies further from the coil. These factors make cerebellar fMRI challenging. They also motivate our approach of combining information across multiple fMRI datasets in an optimal way to benefit from as much high-quality data as possible. To achieve this, our Bayesian Hierarchical framework estimates the reliability of each dataset and weighs the information from each dataset accordingly.

Hence, while these limitations generally affect the quality and interpretability of cerebellar fMRI studies, they do not reduce the utility of fMRI as an atlasing technique when combined with our Hierarchical Bayesian framework.

6. *l142: Greater granularity appears to be better overall - if this is the case, why were higher levels of granularity not tested and/or what is the trade-off that was optimized to choose this level.*

The ability of a parcellation to predict functional boundaries at the individual level indeed increased with higher granularity up to $K=200$ regions (Zhi et al. 2023). However, at the group level, the prediction performance reached its asymptote at around $K=20$ regions. The finer details of the functional organization of the cerebellum therefore reside at the individual level. Our atlas aims to strike a balance between a level of granularity that lends itself as a group prior for individual parcellations (and can take advantage of the granularity achievable at this level) and a succinct and easy to understand summary of regions at the group level. We believe that a hierarchically organized atlas, starting at 68 regions provides a useful and practical compromise.

7. *l231: This section may benefit from some additional context in the beginning as to why the author included this analysis and what it is contributing to the overall picture.*

We have clarified the importance of the connectivity in the results (line 235):

The cerebellum does not work in isolation - indeed, given the uniform cyto-architecture of the cerebellum, functional specialization arises from the different patterns of connectivity (Leiner et al., 1986). We therefore characterized each cerebellar region by determining the areas of the cerebral cortex that most likely provide input to this area.

8. *l296 section: The results here are very compelling, but it may have been interesting to include an analysis combining individual resting state data with the probabilistic group map. Would there be any added value over the group atlas? Does it/would it be expected to muddy the somatomotor regions too much as described earlier? Given the ubiquity of resting state acquisitions, many researchers are likely to be highly interested in this possibility.*

The reviewer is pointing out an interesting extension of this work, since using resting-state data for individual parcellations is likely of great interest to researchers working in developmental or patient populations, where data can only be acquired at rest. We have taken steps towards testing how rest data can be used for individual parcellations. A preliminary analysis of resting-state data from 17 subjects who also participated in the MDTB study suggests that integrating rest data to predict task boundaries is not straightforward: While integrating task data from the individual into our task-based atlas increases prediction performance on held-out task data from the individual, integrating rest data decreases the prediction of that individual's task boundaries. This information loss is particularly prevalent in the motor regions and suggests that the task might differ substantially

from those found at rest in the boundaries that they highlight. We are currently acquiring new data to replicate this in an independent dataset.

9. *l421: The point about potential "anchor tasks" in the discussion is very interesting and potentially of very high importance, additional elaboration here would be useful.*

We used the following tasks within the first session of the MDTB dataset to construct a 10-minute task battery for general localization without specific focus on a particular set of regions or boundaries (for a detailed explanation of each task, please see

<https://multitaskbattery.readthedocs.io/en/latest/instructions.html#task-instructions>):

- NoGo
- Go
- ToM
- VideoAct
- VideoKnots
- UnpleasantScenes
- PleasantScenes
- Math
- DigitJudgement
- CheckerBoard
- SadFaces
- HappyFaces
- IntervalTiming
- MotorImagery
- FingerSimple
- FingerSeq
- Verbal0Back
- Verbal2Back
- Object0Back
- Object2Back
- SpatialImagery
- StroopIncon
- StroopCon
- VerbGen
- WordRead
- VisualSearchSmall
- VisualSearchMed
- VisualSearchLarge
- Rest

Since this type of localizer battery will delineate all atlas regions, it is well-suited for broad research aims that are not specific to a domain, such as tracking the developmental trajectory of functional regions or identifying patterns that differentiate patients from controls.

When targeting specific domains, we advise researchers to consider which regions or region boundaries are most important to delineate in their study and choose the tasks or weigh the time dedicated to the tasks accordingly. For example, in language studies, it might be beneficial to include a set of localizer tasks that can cleanly dissociate regions S1 – S5 from each other. Examples would be the Spatial Imagery and MotorImagery tasks which separate S4-S5 and S1 from other social-linguistic-spatial regions (Supplementary Fig. 5a and b). For these studies, it is also important to dissociate the social-linguistic-spatial regions from the multiple demand regions. This is because the two domains border directly onto each other and are functionally related. The Verb Generation and SpatialMapDiff task would achieve this for the right hemisphere and the AnimatedMovie task for the left hemisphere (Supplementary Fig. 5c and d). A task that highlights the motor regions of speech production, such as the tongue region, should also be performed.

However, even for studies that focus on a particular domain, it is beneficial to include some ‘anchor tasks’ from domains that are not probed. In the case of a language-focussed battery, we would recommend including a task that targets the motor domain (i.e. performing a finger sequence), multiple demand domain (i.e. working memory task) and action domain (i.e. watching videos of people tying a knot). This is to ‘soak up’ any motor, demand or action regions that might show a small overlap with language regions, but do not belong to this domain. An anchor task for each domain would prevent this from happening. To help researchers design and run localizer task batteries for their studies, we have constructed a Python toolbox that can easily run the tasks included into this study and more: <https://multitaskbattery.readthedocs.io/>. A careful evaluation of different Anchor tasks and different types of task batteries, however, is beyond the scope of the paper – we are working on a separate manuscript addressing this issue with new additional multi-task datasets that are currently being collected.

10. Figure 3c: Please comment on the magnitude of the correlations identified between the regions and cerebral cortical networks (which is echoed in the findings in the supplementary materials) - was this expected and how does it compare to previous work?

The values shown in Figure 3c are not correlations, but un-normalized regression coefficients from the regularized multiple regression model. Thus, their magnitude cannot be interpreted, as it depends on the a) standardisation of the data, b) the number of the neocortical areas that are considered, and c) the amount of regularization applied to the model. What is interpretable is the pattern of connectivity, as well as the % variance of the cerebellar data that can be predicted across tasks (Figure 1b).

11. Given the challenge of displaying results on the cerebellum in a meaningful way (both the flatmaps and the volumetric representations each having their own limitations), it would potentially be useful to include, perhaps in the supplementary materials, the atlas rendered in a 3D view of the cerebellar surface from different perspectives.

Thank you for the suggestion, we have added 3D views of the cerebellar surface from different perspectives to the supplementary material to aid the reader in interpreting the maps (see below).

“Figure S5. Atlas in 3D view. Atlas shown at medium granularity (32 regions; 16 per hemisphere) projected onto the pial surface. The central view is showing the posterior side of the cerebellum. The vertically arranged views show the superior side of the cerebellum at the top and the inferior side at the bottom. The horizontally arranged views show the left and right side of the cerebellum.”

Minor points:

12. l15-16: This statement could benefit from added precision. There is substantial work and a reasonable understanding on the contribution of the cerebellum specifically to motor behaviours, it is more it's contribution to cognitive/social behaviours that has been elusive.

We agree with the reviewer that there is substantial work on the contribution of the cerebellum to motor behaviours, and that the function of specific cerebellar regions in specific motor behaviors are well-described. However, we believe that a *general* understanding of the cerebellar contribution to motor control remains elusive. While the predominant conceptualisation of the cerebellum in the motor literature is a forward model, the diversity of inputs and outputs signals of the cerebellum in classical motor tasks is more consistent with a strong diversity of cerebellar contributions even to movement (for a review, see Orban de Xivry & Diedrichsen, 2024). We think it premature to suggest that what the cerebellum contributes to movement and how this contribution is implemented has been resolved. We have therefore refrained from rephrasing the sentence and hope that the reviewer is satisfied with our explanation of the statement.

13. l120-122: *It is not entirely clear what is meant by constraining the atlas to have “spatially symmetric regions across left and right, while allowing different functional profiles”. Please provide some clarification.*

We apologize for the confusion and thank the reviewer for the opportunity to clarify this.

To create a symmetric parcellation, where the boundaries fall into the same place between the left and the right hemisphere, we constrained the voxel assignments to be symmetric. Specifically, we averaged the log-likelihood of a left voxel with its corresponding right voxel within the emission model before integrating it with the group prior in each fitting step. The estimation of functional profiles of each region within the emission model is unaffected by this constraint, enabling the functional profiles of left and right to be independent.

Concretely, this means that two corresponding functional regions on the left and the right can have vastly different responses, even though they have the same boundaries. In the regions that show highly lateralized responses, we see that this is the case: S1R responds strongest to a theory of mind and a verb generation task, which barely elicit any activity in S1L. Meanwhile, S1L activates when watching an animated movie, where S1R is unresponsive. So, despite having landed in the same functional region, S1L and S1R respond differently.

14. l190: *“Executive control” can be a bit of nebulous concept with many possible functions under that umbrella (e.g. inhibition, cognitive flexibility, self-monitoring, etc.) A few words on what this means here would help to clarify.*

The tasks activating the multiple demand regions predominantly stemmed from the Demand dataset (Assem et al., 2022) and probed executive demands of updating, shifting and inhibition. We rephrased the sentence to specify these functions:

“Tasks involving executive control, including updating, shifting and inhibition, consistently activated regions in lobules VI and VII.”

We chose the terminology “Demand regions”, consistent with recent theoretical developments in the study of “executive” function, namely the idea that there is a core network of areas that responds to “multiple demands” (see Duncan et al., 2020).

15. l377: *“It is presently ...” - as written this statement is not entirely clear.*

We have revised this sentence for clarity:

Previous version:

~~“It is presently unclear whether further details will suggest a parsimonious organization or, alternatively, as has evolved in the studies using intensive within-individual mapping, spatial complexity will emerge.”~~

“Future neuroimaging studies might reveal a parsimonious organization or more spatial complexity, as has been suggested by intensive within-individual mapping.”

16. l387: As “Social function” is estimated by very particular tasks this inference might be a little bit of an over generalization. Considering that “social function” is a massive concept, it may be better to be more precise here.

We agree that the term ‘social function’ was not chosen well here and have rephrased the sentence to be clearer on what we mean:

“This work showed that the default network can be divided into two parts, one that is associated with remembering and scene construction (network A), the other that is associated with ~~social function~~ mentalizing (network B).”

17. l396-399: This is fascinating result, but it would be interesting to provide some additional perspectives as to why this might be the case. Is this heterogeneity also seen at the cortical level? Is the author confident that this organization can be so different across individuals, or could it be somehow related to methodological reasons? Also, how confident should we be that the results in an individual are stable over time?, is it possible that instability here may contribute to the observed heterogeneity?

We thank the reviewer for their appreciation of this result and are happy to elaborate on it.

We took a number of precautions to maximise our confidence in the analysis and ensure the results are not driven by methodological factors.

- I. We calculated inter-individual variability voxel-wise rather than for each region, to ensure that differences in region size cannot influence the calculated variability.
- II. We normalized the variability by the reliability to account for any measurement noise. The reliability was calculated by dividing the data of each subject into two parts and correlating the two independent estimates of each subject’s response profile.
- III. To reduce the influence of noise when visualizing the pattern, we averaged each inter-subject correlation value across subjects and divided it by the reliability averaged across subjects to obtain a single value per voxels.

We did not formally compare variability in the cerebellum with variability in the neocortex, but prior work on variability of resting-state networks found that the cerebellum was significantly more variable than the neocortex (Marek et al., 2018).

Finally, we did not analyze the stability of the regions over time in this paper, but we are confident that the finding of greater variability for language and higher-order cognitive regions is not solely driven by time-wise variations. High variability across individuals has been a long-standing finding in the language literature. Despite this inter-individual variability, the spatial pattern of the language network, the degree of lateralization and the extent of language responsiveness are relatively stable within individuals over time (Mahowald et al., 2016; for a review see Fedorenko et al., 2024). It is possible that the extent and precise location of functional regions may change across longer timescales, e.g. childhood development. However, we believe that this did not play a large role in our data, since the age range in the included datasets was restricted to young adults.

18. l410: Here the author mentions that 10 minutes of individual data combined with the group map is better than both the group and individual maps, but in the results (line 323) seem to suggest that 20 minutes is necessary to outperform these. Please clarify.

We thank the reviewer for the opportunity to clarify our result. Combining 10 or 20 minutes of data with the group map numerically outperforms the group and the individual maps in predicting functional boundaries (DCBC) and predicting functional responses (prediction error). When comparing the ability to predict functional boundaries statistically, 10 minutes of individual data sufficed to outperform individual data (DCBC: 10 min combined vs. data $t(23)=9.459$, $p = 2.161e-09$), but not the group atlas (DCBC: 10 min combined vs. group $t(23)=1.539$, $p = 0.137$). For predicting functional responses, 10 minutes of individual data significantly outperformed group and individual data (prediction error: 10 min combined vs. data $t(23)=-9.670$, $p = 1.43e-09$; 10 min combined vs. group $t(23)=-2.469$, $p = 2.14e-02$). To better characterise these results, we amended the text in the discussion:

“...our new atlas offers an alternative, by optimally integrating even limited individual data (~10-20 minutes) with the probabilistic group map.”

19. l421-431: Can the author please provide a brief hypothetical example of what the task battery might look like for a 10-minute localizer?

We have provided an example localizer battery in our response to Comment 9 of Reviewer 2. To ease the adoption of a short task localizer into scanning protocols, we have constructed a Python toolbox where researchers can select a task list and easily run the tasks in the scanner: <https://multitaskbattery.readthedocs.io/>

20. l489: "different dataset" -> datasets

Corrected – thank you!

21. Figure 1b: missing labels for axes

As mentioned in our response to Reviewer 3, comment 4, figure 1b is a multi-dimensional scaling representation of the similarity structure of the single-data set parcellations. This means that both the units of the x- and y-axis are somewhat arbitrary (as is the orientation of the arrangement). In an MDS plot, only the relative distances between the different points are interpretable. We have therefore refrained from adding an axis label.

Figure 3a/c: missing label on colorbar

Corrected – thank you!

Reviewer 2 (Remarks on code availability)

I have not had a chance to review the code.

Reviewer 3

This manuscript describes a new atlas to the already existing body of cerebellar atlases, with the specific advantage of being multi-level and optionally symmetric. The authors also outline an idea of warping individual data based on functional responses, which is interesting and potentially very valuable for cerebellar research. The methodology, data analysis and conclusions are all sound. A significant amount of work is crammed into this manuscript, and as a result the level of detail is not sufficient to really reproduce the presented results without additional information (which is likely to be provided on the authors' website in future).

We thank the reviewer for their positive evaluation and their keen attention to detail in their comments, which helped us improve the readability of the manuscript substantially. We agree with the reviewer that the details provided in this manuscript are likely not sufficient for a full reproduction of the results. We have revisited our figures and text according to the reviewer's specific comments to ensure that all analyses are presented as clearly as possible, within the constrained space of the paper. In case more detail is needed than the paper allows, we kindly refer the reviewer and readers to our code repository containing the scripts and notebooks to reproduce the results and the figures.

- 1. I do wonder how this function-driven alignment fits with the underlying anatomy, as the space of the atlas is still defined by the structural boundaries of the cerebellum. As the authors also point out, the lobular structure does not really provide an appropriate scaffold for functional mapping of the cerebellum, so it is a little confusing that the authors go back to this lobule-defined space to present the atlas in.*

We agree with the reviewer that the cerebellar lobules do not capture functional boundaries in the cerebellum. However, we have chosen to draw lobular boundaries as dashed lines into all functional parcellations presented, because lobular boundaries serve as important anatomical landmarks for understanding how a particular functional region lies in the cerebellum. For example, appreciating how regions S1 and S2 cluster around the horizontal fissure helps the reader place S1 and S2 in the most posterior zones of the cerebellar volume. Since we project all maps into the cerebellar flat map for visualization in this paper, the lobular boundaries are a necessary visual aid to tie the maps in with the 3D volume. Further, lobular boundaries are distinguishable by eye on an MRI scan, which has likely been the reason for their widespread use for dividing the cerebellum. While lobular boundaries are functionally not meaningful, in the absence of functional data, their discernability in a raw, unprocessed MRI scan is a clear advantage for getting a rough estimate of where a functional region would lie in this MRI scan. Finally, lobular boundaries are usually the only reference point in which cerebellar findings in historical papers have been described. We therefore thought it important to link the functional regions in this atlas to this established if outdated way of referencing.

- 2. As the atlas is not dramatically different from the existing MDTB atlas, new functional insights are somewhat limited, this is more the presentation of a new tool for cerebellar imaging.*

We recognize that the substantial differences between this atlas and the existing MDTB parcellation were not sufficiently emphasized in the manuscript. We would like to refer the reviewer to the analysis included in our response to comment 1 of Reviewer 1 which shows the

similarity between the new atlas and the existing MDTB parcellation, as well as the other datasets from which the new atlas was derived. This analysis reveals that the large majority of regions found in our new atlas have highest similarity with regions derived from datasets other than the MDTB. Additionally, we included a more extensive discussion of the novel functional insights provided by this atlas over existing ones in the response to comment 1 of Reviewer 1.

3. *The figures are beautiful, but difficult to read, at least without digging through the text and looking up some of the references, as legends and axis labels are missing in multiple places.*

We are grateful for the reviewer's positive impression of our figures and understand that a lack of clarity in figure legends and labels may have hindered the interpretation of these. We have addressed the reviewer's specific comments on the figures below and carefully implemented the suggested changes to aid the reader.

Specific points:

Introduction

4. *For readability, follow line and order of reasoning of the abstract, or vice-versa.*

Thank you for your suggestion. We have amended the abstract to follow the line and order of reasoning in the introduction for readability:

“The human cerebellum is activated by a wide variety of cognitive and motor tasks. Previous functional atlases have relied on single task-based or resting-state fMRI datasets. Here, we present a functional atlas that integrates information from 7 large-scale datasets, outperforming existing group atlases. The new atlas has three further advantages: **Finally, First, the atlas allows for precision mapping in individuals: The integration of the probabilistic group atlas with an individual localizer scan results in a marked improvement in prediction of individual boundaries.** Second, we provide both asymmetric and symmetric versions of the atlas. The symmetric version, which is obtained by constraining the boundaries to be the same across hemispheres, is especially useful in studying functional lateralization. **First, Finally, the regions are hierarchically organized across 3 levels, allowing analyses at the appropriate level of granularity.** Overall, the new atlas is an important resource for the study of the interdigitated functional organization of the human cerebellum in health and disease.”

Results

Generally, the reader is not guided very much through the figures. The points below are just questions that occurred to me while reading. I suspect that many are answered in ref (8), so I put these points here mostly to help make the current manuscript independently readable.

Figure 1 starts with a map labelled 'MDTB'. However, this is not the atlas that the same lab has been sharing as the one generated from the MDTB database

(<https://www.diedrichsenlab.org/imaging/mdtb.htm>). Is this a modified version?

The reviewer is correct in that the MDTB parcellation in Figure 1a is not the same as shared in the King et al., 2019 paper. For a fair comparison between a parcellation learned from multiple datasets and single-dataset parcellations (including a parcellation based on the MDTB dataset

only), we thought it important to use the same algorithm to derive all parcellations. We thereby ensured that any advantage of the fused map would be due to the combined information across several datasets and not simply due to a more advanced parcellation algorithm. Since the MDTB parcellation with 7, 10 and 17 regions from 2019 was generated using convex semi-non-negative vector factorization, we therefore used our Hierarchical Bayesian framework to derive a new parcellation from the MDTB dataset, which can be seen in Figure 1a. We also base the comparison on the average performance across several granularities (10, 20, 34, 40 and 68 regions), which for the existing MDTB parcellations derived with semi-non-negative vector factorization does not exist. However, when comparing our final atlas map to existing parcellations, we indeed use the 2019 map at 7, 10 and 17 regions (Supplementary Fig. S1).

1. *Figure 1a is lacking a legend. Do same colours in the different maps refer to the same networks?*

As mentioned in the response to Reviewer 1, Comment 2, the assignment of numbers to parcels is not meaningful in Figure 1a (and there is no 1:1 correspondence of parcels across maps). We have therefore refrained from adding a key with parcel numbers to this figure to avoid implying some correspondence across maps that does not exist.

We have, however, added a brief explanation regarding the colour scheme to the legend of figure 1a to clarify this point:

“Figure 1. Building a functional atlas of the cerebellum across datasets. a. Parcellations (K=68) derived from each single dataset. The probabilistic parcellation is shown as a winner-take-all projection onto a flattened representation of the cerebellum. **Functionally similar regions are colored similarly within a parcellation (see methods: parcel similarity). Across parcellations we rotated the color assignment using three spatial anchor points in multi-demand, motor, and social linguistic regions. [...]**”

2. *The sentence ‘A strong boundary between ...’ (ln 67) should be supported by some visual guidance in the figure. How can the reader see that ‘the somatotopic dataset did not delineate cognitive regions in lobule VII well’ ?*

We regret the confusion and have amended the text in the paragraph beginning in line 67 to give the reader better guidance on which boundaries we refer to:

“A ~~clear~~ **smooth** boundary between motor regions in lobule I-VI and cognitive regions in lobule VII was present in all parcellations (e.g. **between the magenta and pink regions in MDTB and Demand dataset in lobule VI.**)”

“For example, the somatotopic dataset only tested individual body movements, and therefore resulted in a clear somatomotor map, but did not delineate cognitive regions in lobule VII well, **as can be seen by the fragmented pattern in Crus I/II and lobule IX.**”

3. *Line 73: consistent boundaries – consistent with what?*

We have edited the text for clarity:

“Parcellations based on resting-state data (HCP) showed consistent boundaries in regions related to the default network (lobules VII) but appear to delineate other regions (e.g. motor) less finely.”

4. *Figure 1b is missing labels on the x and y axis. What information is provided here?*

Figure 1b is a multi-dimensional scaling representation of the similarity structure of the single-data set parcellations. That is, both the units of the x- and y-axis are somewhat arbitrary (as is the orientation of the arrangement). In an MDS plot, only the relative distances between the different points are interpretable.

5. *Line 92: were significantly more similar to the MDTB*

Corrected – thank you!

6. *Is the figure reference on line 128 correct? The text jumps unexpectedly from Figure 1 to 4 here.*

The figure reference was incorrect, but we have corrected it now – thank you for spotting this!

7. *Is the line of missing voxels in Figure 4e the symmetry axis? I would have expected this to be straight.*

The reviewer has correctly spotted the symmetry axis, showing up as missing voxels (NaN values) in the boundary symmetry plot, since boundary symmetry cannot be calculated there. It does not appear straight in the flat map projection, because we did not create a symmetric flat map but used the existing, asymmetric flat map (Diedrichsen & Zotow, 2015). While the missing voxels therefore run ‘straight’ across the midline of the cerebellum in the volume, the projection to the cerebellar flat map surface does not retain this symmetry.

8. *Line 157: its*

Corrected.

9. *The term ‘action’ as opposed to ‘motor’ is not completely intuitive, at least not to me.*

We thank the reviewer for the opportunity to clarify what the ‘action’ regions are. The term ‘action regions’ was used in this manuscript to refer to regions involved in action observation. The action observation regions are functionally related to the motor regions and activate when performing a motor task. However, they can be distinguished from motor regions by their strong response to action observation tasks, such as watching a knot-tying video, where motor regions show no activity (see Supplementary Fig. S2).

10. *Figure 3 misses titles for the subpanels. As ‘Fusion’ is included with an error bar in panel b, it is not clear why it is left out of panel a. Why was the MTBD-Highres not included here?*

The MDTB-Highres data set was not included in the connectivity model, as only the posterior aspect of the cerebrum was covered in data acquisition. This prevented us from fitting a comparable effective connectivity model between neocortex and cerebellum.

We apologize for the oversight and have added the ‘Fusion’ model to all relevant plots and titles for the subpanels.

11. *Line 280: Additionally, it is Corrected.*

12. *Figure 5 – panel (a) lacks a legend.*

We chose not to include a legend for the repeated presentation of the atlas at medium granularity (32 regions) in Figures 4a, 4b and 5a, as this would distract from the main points of the figures. However, we have added subheadings for each of the subpanels and amended the legend of Figure 5a to clarify what is shown:

“The region colors correspond to the atlas at medium granularity (32 regions).”

13. *Does it matter which 20 or 10 min of individual data are used to estimate the individuals’ map? Presumably some combinations of tasks are more useful than others. A reference to the methods or short description of a suggested localiser task set would be useful for replication.*

The 10, 20, ... minutes to individualize the parcellations were from the first session in the MDTB dataset, which includes all tasks of task set A (King et al., 2019). Therefore, adding another 10 minutes of data to the estimation of individual regions will only add more data, but not more tasks. The choice of which 10-20 minutes are used for estimation will therefore not affect the delineation of regions, since it will consist of the same tasks. We have listed the tasks included into each 10-minute run in our response to Comment 9 of Reviewer 2. For a more detailed explanation of each task, we would like to refer the reviewer to the task instructions along with screenshots of the task we have released as part of our documentation of our MultiTaskBattery toolbox for running these tasks in the scanner:

<https://multitaskbattery.readthedocs.io/en/latest/instructions.html#task-instructions>)

14. *The individual data used here is drawn from the group of participants the atlas was based on. The size of the databases is perhaps sufficient to avoid circularity, but would the mapping also work for new participants, e.g. scanned on a different scanner (location/vendor/field strength)?*

Since submission, we have acquired an independent dataset of 7 participants to replicate our findings. All participants performed 8 runs of a 10-minute localizer battery that consisted of 7 tasks used in the first session of the MDTB dataset and 7 novel localizer tasks (Fedorenko et al., 2010; Scott et al., 2017; Saadon-Grosman et al., 2022; Buckner et al., 2023). The data was acquired on a different scanner and at a different field strength (7T, as opposed to the 3T MDTB data). We matched pre-processing steps and data extraction to the steps in this manuscript.

Using this independent dataset, we replicate our finding that the integration of individual data substantially improves the prediction of functional boundaries. We split the 8 runs in half and used the first four runs to individualize the atlas regions. We then compared the ability of these individualized regions in their ability to predict functional boundaries in the held-out data of the same subject to the ability of the group atlas. We find that the individualized boundaries significantly outperform the group atlas ($t(6) = -7.571$, $p = 2.76e-04$)

15. Line 405: “predict individual functional data better than the group map.” (there are similar sentences elsewhere in the text). What does the term ‘better’ reflect here? Higher DCBC values?

Indeed, higher DCBC values indicate better prediction of functional boundaries. We realize that the DCBC interpretation was mentioned solely in the methods and have therefore added this information into the main text where the DCBC is mentioned:

“This ability was quantified using the Distance-Controlled Boundary Coefficient (DCBC) which compares the correlation between within-parcel voxel-pairs to the correlation between voxels-pairs across a boundary, while controlling for spatial distance, with higher values indicating better performance.”

“We then evaluated these parcellations on how well they separated functional regions (DCBC, higher DCBC indicating better separation; Fig. 5d) and predicted the functional profiles (prediction error, lower error indicating better prediction; Fig. 5e).”

16. *Line 557: to compare the the similarity*

Corrected – thank you!

17. *Line 587: grammar? Please check*

We noticed two small typos, thank you for spotting them. The sentence now reads:

“To assess the ability of a given parcellation to predict functional responses in individual held-out data, we calculated a prediction error.”

References

- King, M. et al. (2019) 'Functional boundaries in the human cerebellum revealed by a multi-domain task battery', *Nature Neuroscience*, 22(8), pp. 1371–1378. Available at: <https://doi.org/10.1038/s41593-019-0436-x>.
- Mahowald, K. and Fedorenko, E. (2016) 'Reliable individual-level neural markers of high-level language processing: A necessary precursor for relating neural variability to behavioral and genetic variability', *NeuroImage*, 139, pp. 74–93. Available at: <https://doi.org/10.1016/j.neuroimage.2016.05.073>.
- Fedorenko, E., Ivanova, A.A. and Regev, T.I. (no date) 'The language network as a natural kind within the broader landscape of the human brain', *Nature Reviews Neuroscience* [Preprint]. Available at: <https://doi.org/10.1038/s41583-024-00802-4>.
- Sereno, M.I. et al. (2020) 'The human cerebellum has almost 80% of the surface area of the neocortex', *Proceedings of the National Academy of Sciences of the United States of America*, 117(32), pp. 19538–19543. Available at: <https://doi.org/10.1073/pnas.2002896117>.
- Brooks, J.C.W. et al. (2013) 'Physiological noise in brainstem fMRI', *Frontiers in Human Neuroscience*, 7(OCT), pp. 1–13. Available at: <https://doi.org/10.3389/fnhum.2013.00623>.
- Assem, M. et al. (2024) 'Basis of executive functions in fine-grained architecture of cortical and subcortical human brain networks', *Cerebral Cortex* [Preprint]. Available at: <https://doi.org/10.1093/cercor/bhad537>.
- Leiner, H.C., Leiner, A.L. and Dow, R.S. (1986) 'Does the Cerebellum Contribute to Mental Skills?', *Behavioral Neuroscience* [Preprint]. Available at: <https://doi.org/10.1037/0735-7044.100.4.443>.
- Duncan, J., Assem, M. and Shashidhara, S. (2021) 'Europe PMC Funders Group Integrated intelligence from distributed brain activity', 24(10), pp. 838–852. Available at: <https://doi.org/10.1016/j.tics.2020.06.012.Integrated>.
- Marek, S. et al. (2018) 'Spatial and Temporal Organization of the Individual Human Cerebellum', *Neuron*, 100(4), pp. 977-993.e7. Available at: <https://doi.org/10.1016/j.neuron.2018.10.010>.
- Fedorenko, E. et al. (2010) 'New method for fMRI investigations of language: Defining ROIs functionally in individual subjects', *Journal of Neurophysiology*, 104(2), pp. 1177–1194. Available at: <https://doi.org/10.1152/jn.00032.2010>.
- Saadon-Grosman, N. et al. (2022) 'A Third Somatomotor Representation in the Human Cerebellum', *Journal of Neurophysiology*, (April), pp. 1051–1073. Available at: <https://doi.org/10.1152/jn.00165.2022>.
- Scott, T.L., Gallée, J. and Fedorenko, E. (2017) 'A new fun and robust version of an fMRI localizer for the frontotemporal language system', *Cognitive Neuroscience* [Preprint]. Available at: <https://doi.org/10.1080/17588928.2016.1201466>.

Wynn, J.K. et al. (2015) 'Impaired target detection in schizophrenia and the ventral attentional network: Findings from a joint event-related potential-functional MRI analysis: Target stimulus ERP/fMRI analysis in schizophrenia', *NeuroImage: Clinical* [Preprint]. Available at: <https://doi.org/10.1016/j.nicl.2015.07.004>.

Wang, Z. et al. (2024) 'Intrinsic structural covariation links cerebellum subregions to the cerebral cortex', *bioRxiv*, pp. 1–44.

Xivry, J.O. De and Diedrichsen, J. (2024) 'cerebellum suggests a diversity of function ☆', *Current Opinion in Behavioral Sciences*, 57, p. 101386. Available at: <https://doi.org/10.1016/j.cobeha.2024.101386>.

Diedrichsen, J. and Zotow, E. (2015) 'Surface-based display of volume-averaged cerebellar imaging data', *PLoS ONE*, 10(7), pp. 1–18. Available at: <https://doi.org/10.1371/journal.pone.0133402>.

Reviewer #1 (Remarks to the Author):

The authors have addressed my concerns.

Reviewer #2 (Remarks to the Author):

I am satisfied with the changes that the authors have made.

5. If there is room within the discussion, some of the response to point 5. (reviewer 2) may also be useful to include. Though perhaps not completely critical to the current work, it may help to highlight the gains that the approach can provide.

Reviewer #3 (Remarks to the Author):

The authors have addressed all my comments in the reply to reviewers and made appropriate changes in the new version of the manuscript.

While I understand that this version of the atlas might be considered an updated or better version of the MDTB-atlas, as a future user of these atlases I'm still a bit confused as to when one should be used or the other. Or do the authors feel the MDTB is simply not relevant anymore now that this new one exists?

Response to Reviewers

We thank the reviewers and editor for their positive assessment and their helpful and constructive comments.

Reviewer 1

The authors have addressed my concerns.

We would like to thank the reviewer for constructive review of our manuscript.

Reviewer 2

I am satisfied with the changes that the authors have made.

5. If there is room within the discussion, some of the response to point 5. (reviewer 2) may also be useful to include. Though perhaps not completely critical to the current work, it may help to highlight the gains that the approach can provide.

We would like to thank the reviewer for their constructive review of our manuscript, and we are pleased the Reviewer is satisfied with the changes we have made to the manuscript. We have opted to leave the response to point 5 out of the main manuscript text, as it would warrant discussion that we feel is beyond the scope and space limits of the manuscript. However, since the responses will be published alongside the manuscript, we are confident that the interested reader will benefit from this additional information in the published response.

Reviewer 2

The authors have addressed all my comments in the reply to reviewers and made appropriate changes in the new version of the manuscript.

While I understand that this version of the atlas might be considered an updated or better version of the MDTB-atlas, as a future user of these atlases I'm still a bit confused as to when one should be used or the other. Or do the authors feel the MDTB is simply not relevant anymore now that this new one exists?

We would like to thank the reviewer for their positive assessment and constructive review of our manuscript.

We apologize for the confusion. We are of the opinion that the atlas should supersede the MDTB atlas, as it includes all the information contained in the former. Additionally it is more generalizable and has greater utility (offering probabilistic maps suited for individual precision mapping, a nested hierarchy of parcellations, a symmetric version, and connectivity maps).